# Rate-limiting transport of positively charged arginine residues through the Sec-machinery is integral to the mechanism of protein secretion

William J Allen[1]*, Robin A Corey[1†], Daniel W Watkins[1], A Sofia F Oliveira[1,2], Kiel Hards[3], Gregory M Cook[3], Ian Collinson[1]*

[1]School of Biochemistry, University of Bristol, University Walk, Bristol, United Kingdom; [2]School of Chemistry, University of Bristol, University Walk, Bristol, United Kingdom; [3]Department of Microbiology and Immunology, University of Otago, Dunedin, New Zealand

*For correspondence:
w.allen@bristol.ac.uk (WJA);
ian.collinson@bristol.ac.uk (IC)

Present address: [†]Department of Biochemistry, University of Oxford, Oxford, United Kingdom

**Competing interest:** The authors declare that no competing interests exist.

**Abstract** Transport of proteins across and into membranes is a fundamental biological process with the vast majority being conducted by the ubiquitous Sec machinery. In bacteria, this is usually achieved when the SecY-complex engages the cytosolic ATPase SecA (secretion) or translating ribosomes (insertion). Great strides have been made towards understanding the mechanism of protein translocation. Yet, important questions remain – notably, the nature of the individual steps that constitute transport, and how the proton-motive force (PMF) across the plasma membrane contributes. Here, we apply a recently developed high-resolution protein transport assay to explore these questions. We find that pre-protein transport is limited primarily by the diffusion of arginine residues across the membrane, particularly in the context of bulky hydrophobic sequences. This specific effect of arginine, caused by its positive charge, is mitigated for lysine which can be deprotonated and transported across the membrane in its neutral form. These observations have interesting implications for the mechanism of protein secretion, suggesting a simple mechanism through which the PMF can aid transport by enabling a 'proton ratchet', wherein re-protonation of exiting lysine residues prevents channel re-entry, biasing transport in the outward direction.

## Editor's evaluation

Using a novel bioluminescence-based assay combined with rigorous kinetic modeling, Collinson and colleagues dissect the sequence features of client proteins that influence SecA/SecYEG-mediated protein translocation across the bacterial inner membranes. This study pushes the description of this important cellular pathway towards a highly detailed level, which will potentially advance our understanding of ATP-driven protein secretion mechanisms in bacteria.

## Introduction

Secretion of proteins synthesised in the cytosol is governed by cleavable N-terminal signal sequences (SS), necessary and sufficient for targeting (***Blobel and Dobberstein, 1975***), possibly augmented by information in the mature protein such as a lowered propensity to fold and exposed hydrophobic patches (***Chatzi et al., 2017***). Secretory and membrane proteins are recognised by factors for delivery, usually in an unfolded state, to bespoke protein translocation machineries (translocons) for carriage across or into cellular membranes. Amongst these, and responsible for the majority both of protein

**Figure 1.** The Brownian ratchet model. Mechanism of pre-protein transport, based on *Allen et al., 2016*. Cycles of ATP binding and hydrolysis by SecA (blue) allow blocks (purple) in the pre-protein substrate (green) to diffuse outwards through SecYEG (red) one at a time, but backsliding is prevented, leading to directional transport. See text for more detail.

The online version of this article includes the following figure supplement(s) for figure 1:

**Figure supplement 1.** The NanoLuc pre-protein transport assay.

secretion and membrane protein insertion, is the universally conserved Sec system: at its core the heterotrimer SecYEG/β in plasma membrane of bacteria and Archaea or Sec61αβγ in the endoplasmic reticulum of eukaryotes. The Sec system transports unfolded proteins, either immediately as they emerge from the ribosome (co-translationally) or following their release (post-translationally; *Arkowitz et al., 1993*).

In bacteria, membrane protein insertion is generally co-translational, whereas almost all secretion is post-translational, mediated by the cytosolic ATPase SecA (*Lill et al., 1990*). Prior to passage across the membrane, pre-proteins with a SS are recognised by SecA, either directly or with the assistance of chaperones such as SecB, and brought to SecYEG in the membrane (*Figure 1*; *Hartl et al., 1990*; *Kumamoto and Beckwith, 1983*; *Oliver and Beckwith, 1982*). The SS unlocks the translocon and initiates transport by inserting as a hairpin through the channel (step *i* in *Figure 1*; *Corey et al., 2016b*; *Ma et al., 2019*), using ATP hydrolysis by SecA as an energy source (*Economou and Wickner, 1994*; *Fessl et al., 2018*; *Lill et al., 1990*). Transport of the rest of the of the polypeptide substrate proceeds stepwise (steps *ii-v*, elaborated below), with cycles of ATP turnover (*Brundage et al., 1990*), aided in vivo by the electrochemical gradient of protons across the membrane – the proton-motive force (PMF; positive and acidic outside; *Schiebel et al., 1991*). Once the pre-protein has crossed the membrane, the SS is cleaved off by signal peptidase and the transported protein released (*Josefsson and Randall, 1981*); this release step appears to require additional factors – most likely the periplasmic chaperone PpiD (*Antonoaea et al., 2008*; *Fürst et al., 2018*) – as it is very slow when measured in vitro using ex vivo or purified and reconstituted components (*Allen et al., 2020*; *Mao et al., 2020*).

Results from our laboratory and elsewhere, using the model *Escherichia coli* Sec system, have shown how ATP turnover could be coupled to directional pre-protein movement (*Allen et al., 2016*; *Catipovic et al., 2019*; *Catipovic and Rapoport, 2020*; *Corey et al., 2019*). Adding a non-hydrolysable ATP analogue to the system, which emulates the effect of nucleotide exchange (step *ii* in *Figure 1*) has two major observable effects: it widens the channel through SecYEG (*Allen et al., 2016*) while tightening the clamp around pre-protein in SecA (*Ahdash et al., 2019*; *Catipovic et al., 2019*). Nucleotide exchange itself is promoted by perturbation of the two-helix finger (2HF; *Allen et al., 2016*; *Zimmer et al., 2008*) of SecA caused by the presence of pre-protein at the entrance to the channel through SecYEG (*Allen et al., 2016*). Together, these observations led us to propose a model

in which pre-protein moves through the channel primarily by diffusion (*Allen et al., 2016*). Sequences that cannot diffuse across the membrane in the ADP-bound state (blocks, purple in *Figure 1*) trigger nucleotide exchange (step *ii*), opening the SecYEG channel to allow them to slide through (step *iii*) and simultaneously clamping SecA shut to prevent them slipping backwards. ATP hydrolysis resets the channel (step *iv*), trapping block sequences that have diffused across on the outside of the membrane, thus providing directionality. This process (steps *ii-iv*) is repeated for each remaining block (step *v*) until the entire pre-protein has crossed the membrane.

A major prediction of the above model is that not every ATP turnover will give rise to a transport event, as step *ii* is reversible. Recent high-resolution protein transport data, collected using a new assay based on split NanoLuc luciferase (*Dixon et al., 2016*; *Pereira et al., 2019*), lent support to this notion. In this assay, the large fragment of NanoLuc (11S) is encapsulated within proteoliposomes (PLs) or inverted membrane vesicles (IMVs) containing SecYEG, while a high-affinity complementary small fragment (pep86) is incorporated into the pre-protein (*Figure 1—figure supplement 1*). As soon as pep86 enters the vesicle it combines with 11S, producing a luminescence signal (*Figure 1—figure supplement 1*; *Allen et al., 2020*; *Pereira et al., 2019*). Using detailed kinetic modelling, we showed that transport occurs in about five steps for the model 161 amino acid pre-protein pSpy, and that each step requires many ATP turnovers to resolve (an estimated 120 for pSpy in vitro, see *Allen et al., 2020*). However, we were unable to show exactly what these steps – presumably the blocks in the above model – physically correspond to. It also remains unclear how the PMF contributes.

To answer these questions, we have now generated a range of different pSpy variants with different physical and chemical properties. Employing the NanoLuc assay to measure their transport both in the presence and absence of proton-motive force (PMF), we show that transporting positively charged residues across the membrane is the slowest step of transport, with bulkier residues also increasing secretion time. This strong barrier to positive charges is partially overcome by deprotonating lysines at the cytosolic face of the membrane; for this reason, arginines, which have a much higher p$K_a$ than lysine, are by far the hardest amino acid to transport. Surprisingly, however, charged residues seem unaffected by the electrical component of PMF ($\Delta \phi$) in our experimental setup – even though electrophoresis ought to be favourable, given that deprotonation of lysines confers a net negative charge to the polypeptide as it passes though the channel towards the positive exterior.

We also find that removing all arginine residues from the pre-protein hugely increases the number of kinetic transport steps, despite speeding up transport overall. Variable step size is characteristic of diffusion-based model such as the one presented above (*Figure 1*). For a power stroke mechanism, on the other hand, for example where an ATP-dependent piston movement of SecA pushes stretches of polypeptide across the membrane (*Catipovic et al., 2019*; *Catipovic and Rapoport, 2020*) step size depends on SecA, and should thus be invariant. Taken together, our results provide important new mechanistic insights of the mechanism of protein secretion; both the insights and the experimental approaches are potentially applicable to other machineries that transport unfolded proteins.

## Results

### The chemical and physical properties of pre-protein affect its transport characteristics

We started by investigating which physical properties of a pre-protein determine how fast it is transported through SecYEG, by systematically varying the amino acid sequence of the model Sec substrate pSpy. It quickly became apparent that altering pSpy affects its solubility in transport buffer, potentially along with other properties such as affinity for SecA, and the rates of initiation and termination. We therefore created constructs consisting of the pSpy SS followed by three tandem mature (m)Spys, with a pep86 sequence after the second (pSpy$_{XLX}$; XLX in *Figure 2a*) – and altered only the central one. The two flanking native Spy sequences ensure that the beginning and end of transport are always the same, and prevent the less stable Spy variants from precipitating upon dilution out of urea. As a control, we created the same construct but with pep86 after the first mSpy (pSpy$_{LXX}$, LXX in *Figure 2a*). The difference in transport time between these two proteins (measured from the lag before transport signal appears, see *Figure 2b* and *Allen et al., 2020*) corresponds exactly to the time it takes to transport the central mSpy (pink in *Figure 2a and c*). The lags for these match perfectly with those for the

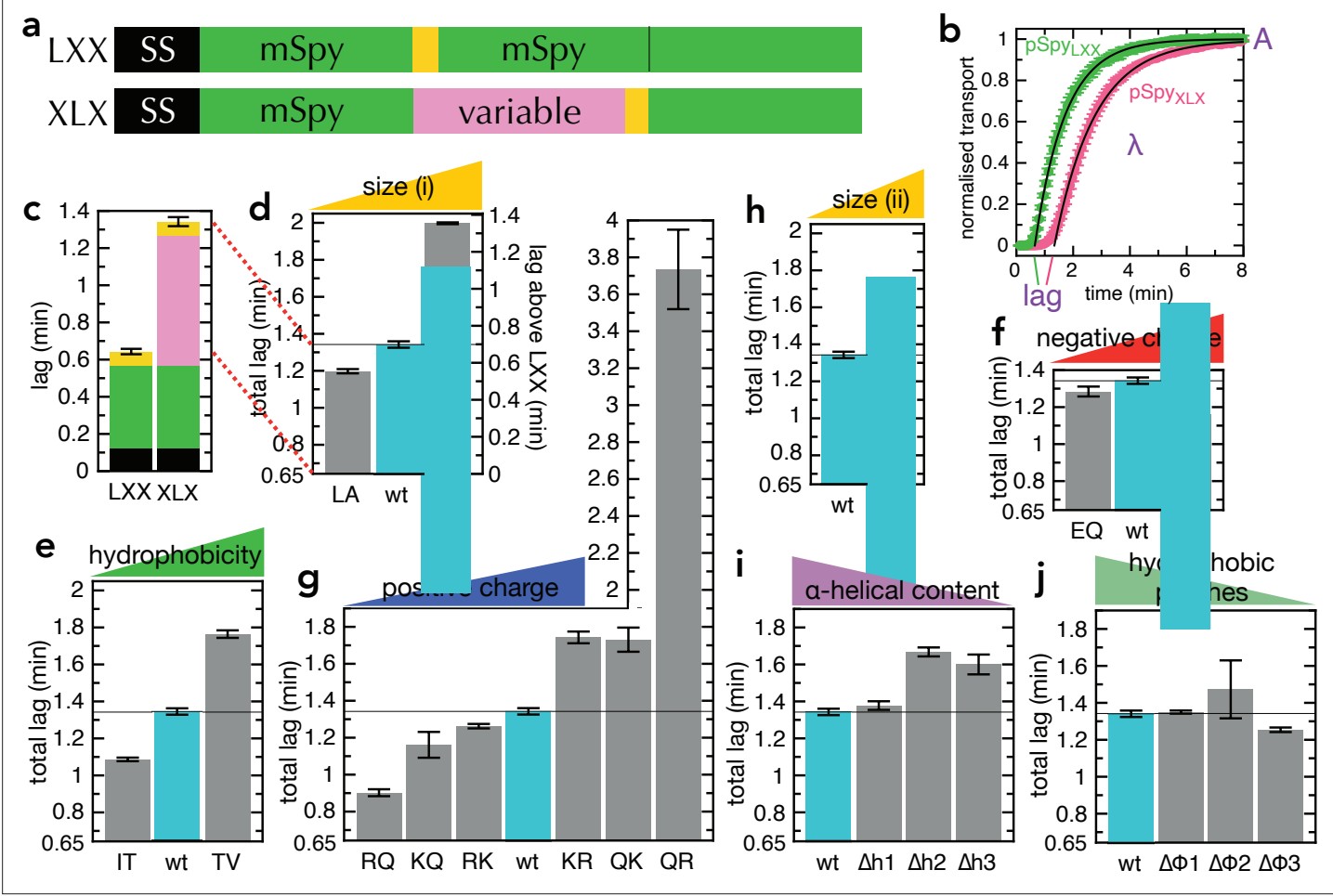

**Figure 2.** Transport of pSpy$_{XLX}$ variants. (**a**) Schematic of pSpy$_{LXX}$ (LXX) and pSpy$_{XLX}$ (XLX). Transport occurs from the N- to C-terminus (left to right), and a luminescence signal appears as soon as pep86 (yellow) enters the lumen of the PL. In these constructs 'X' refers to mSpy with no pep86, and 'L' refers to mSpy followed by a functional (Light) pep86. (**b**) Transport traces of pSpy$_{LXX}$ (green) and pSpy$_{XLX}$ (pink), normalised to give an amplitude (A; signal when the transport reaction reaches completion) of 1, and fitted to the simple lag +single exponential model (**Allen et al., 2020**). In this model, lag is the minimum time required for pre-protein transport, and corresponds to the sum of time constants for all transport steps (equal to 1/$k$, where $k$ is the rate constant for that step). $\lambda$ is a complicated variable, incorporating the transport rates but also the probability of transport pausing or failure and resetting (see **Allen et al., 2020** for more details). Data are the average and SEM from three (pSpy$_{LXX}$) or 11 (pSpy$_{XLX}$) experimental replicates. (**c**) Lag (taken from panel **b**) for pSpy$_{LXX}$ and pSpy$_{XLX}$. (**d-j**) Lags for a range of pSpy$_{XLX}$ variants, where the central mSpy has its chemical and physical properties varied (see text for details; wt (wild type) is native Spy). In each case, the y-axis starts at the transport time for pSpy$_{LXX}$, so the visible part of the bar corresponds to the transport time only of the variable region. Data show the average and SEM from three experimental replicates. Almost all the variants are statistically significant from wt; a table of p-values is included in **Supplementary file 3a**.

The online version of this article includes the following source data and figure supplement(s) for figure 2:

**Source data 1.** Raw data for **Figure 2** and **Figure 3**.

**Figure supplement 1.** Comparison of new and old transport substrates.

**Figure supplement 2.** Amino acid and sequence properties.

pSpy$_{4x}$ series used previously (four tandem Spys; **Allen et al., 2020**; **Figure 2—figure supplement 1a-b**), confirming that lag is indeed a true and sensitive measure of transport time.

Our first set of pSpy$_{XLX}$ variants were created by replacing 6–8 of one type of residue with another, distributed as evenly as possible through mSpy (all sequences shown in **Supplementary file 1**). We chose four general amino acid properties: size (L→A, A→L; **Figure 2d**), hydrophobicity (T→V, I→T; **Figure 2e**), negative charge (Q→E, E→Q; **Figure 2f**), and positive charge (R→Q, Q→R, K→Q, Q→K, K→R, R→K; **Figure 2g**). In the case of positive charge, we looked both at lysine – which can in some environments be deprotonated at physiological pH (**Isom et al., 2011**) – and arginine, which is

generally considered to always retain its positive charge (**Harms et al., 2011** see also below). We find that transport is slower (longer lag than native pSpy$_{XLX}$ above that of pSpy$_{LXX}$, 0.65 min) when more bulky, hydrophobic and positively charged residues are present, while negative charges have limited effect. By far, the strongest effect comes from changing the number of arginines: removing all eight (RQ in **Figure 2g**) reduces the transport time threefold (from 0.64 min to 0.26 min), while adding a further eight (QR in **Figure 2g**) slows transport over fourfold (to 3.1 min).

Leucine is more hydrophobic than alanine, while threonine is somewhat smaller than isoleucine or valine (**Figure 2—figure supplement 2a**), so these experiments do not distinguish particularly effectively between bulkiness and hydrophobicity. We therefore designed an additional variant in which all five phenylalanine residues were replaced with tryptophan (F→W; **Figure 2h**). This increases bulkiness while decreasing hydrophobicity according to most hydrophobicity scales (**Figure 2—figure supplement 2a-d**), in theory allowing the two effects to be disentangled. The results reveal that the F→W substitutions slow transport, which suggests that residue size is a more important factor than hydrophobicity. It should be noted, however, that tryptophan is more hydrophobic than phenylalanine according to the Wimley-White scale (**Wimley and White, 1996**); hence, this substitution is not an absolute confirmation of the importance of size over hydrophobicity.

We have previously shown that the ATP hydrolytic cycle of SecA can influence the formation of secondary structure (primarily α-helix) within the translocating pre-protein (**Corey et al., 2019**), promoting its unfolding at the cytosolic entrance to SecYEG and formation on the periplasmic side. To explore the effect of this on transport kinetics, we designed pSpy$_{XLX}$ variants in which the helical propensity of one, two or three regions was reduced, without affecting the hydrophobic character of that region (Δh1, Δh2 and Δh3; **Figure 2i**; **Figure 2—figure supplement 2e**, purple). A converse set of mutations, in which the hydrophobic character is reduced without altering the helical propensity, was also generated (Δφ1, Δφ2 and Δφ3; **Figure 2j**; **Figure 2—figure supplement 2e**, green). The transport parameters with these variants show that if sufficient helical content is removed it somewhat slows transport (**Figure 2i**), supporting the notion that helix formation is part of the mechanism of transport (**Corey et al., 2019**). Removing hydrophobic patches, meanwhile, has marginal if any effect on transport rate (**Figure 2j**); taken together with the other results above, this suggests residue size is a bigger factor than hydrophobicity in determining transport rate.

## Specific measurement of the transport steps for the pSpy$_{XLX}$ variants

The lag is a useful measure of overall transport time, but it cannot distinguish between a large number of fast steps and a small number of slow steps. For this, we employed a numerical model of transport, derived previously and implemented in Berkeley Madonna (**Figure 3—figure supplement 1a**; **Allen et al., 2020** see Materials and methods). In this model, binding of pre-protein to the Sec system (with on and off rates $k_{on}$ and $k_{off}$) starts at equilibrium. Addition of ATP starts the reaction, allowing initiation ($k_{init}$, equal to $k_{step}$ and corresponding to step $i$ in **Figure 1**) followed by n kinetic transport steps (each one equivalent to steps $ii$-$iv$ in **Figure 1**) with the rate $k_{step}$. The final step gives rise to a luminescent product, as NanoLuc formation is essentially instant on the time scale of transport (**Allen et al., 2020**). Two additional rate constants are required to describe the data fully: $k_{fail}$, the rate at which translocating pre-protein dissociates and transport can be restarted; and $k_{block}$, where the pre-protein becomes permanently trapped in the channel, preventing further transport at that site. A final parameter, 'brightness', simply represents the amount of luminescent signal per NanoLuc, normalised to give a maximum signal of 1.

Just as described previously (**Allen et al., 2020**), best fits to the experimental data were calculated using the Curve Fit function of Berkeley Madonna over a range of values for n, allowing $k_{step}$, $k_{fail}$ and brightness to vary but fixing the other rate constants to previously determined values. The best fit is taken as the one with the lowest root mean square deviation (RMSD) between the model and the data, normalised to 1 to allow comparison between different data sets. Using this analysis, we find that transport of pSpy$_{LXX}$ is best described by 6 steps with an average $k_{step}$ of 5.30 min$^{-1}$ (green in **Figure 3a**), while transport of pSpy$_{XLX}$ proceeds in 10 steps with $k_{step}$ = 4.42 min$^{-1}$ (pink in **Figure 3a**).

To extract n specifically for transport of the second mSpy of pSpy$_{XLX}$ (**Figure 2a**, pink; the variable one in the above constructs; n$_{var}$), we split the Berkeley Madonna mathematical model into two parts (**Figure 3—figure supplement 1b**) – transport of the 'native' section and the 'variable' section. All parameters for transport of the native mSpy (including initiation and pep86 transport) were fixed to

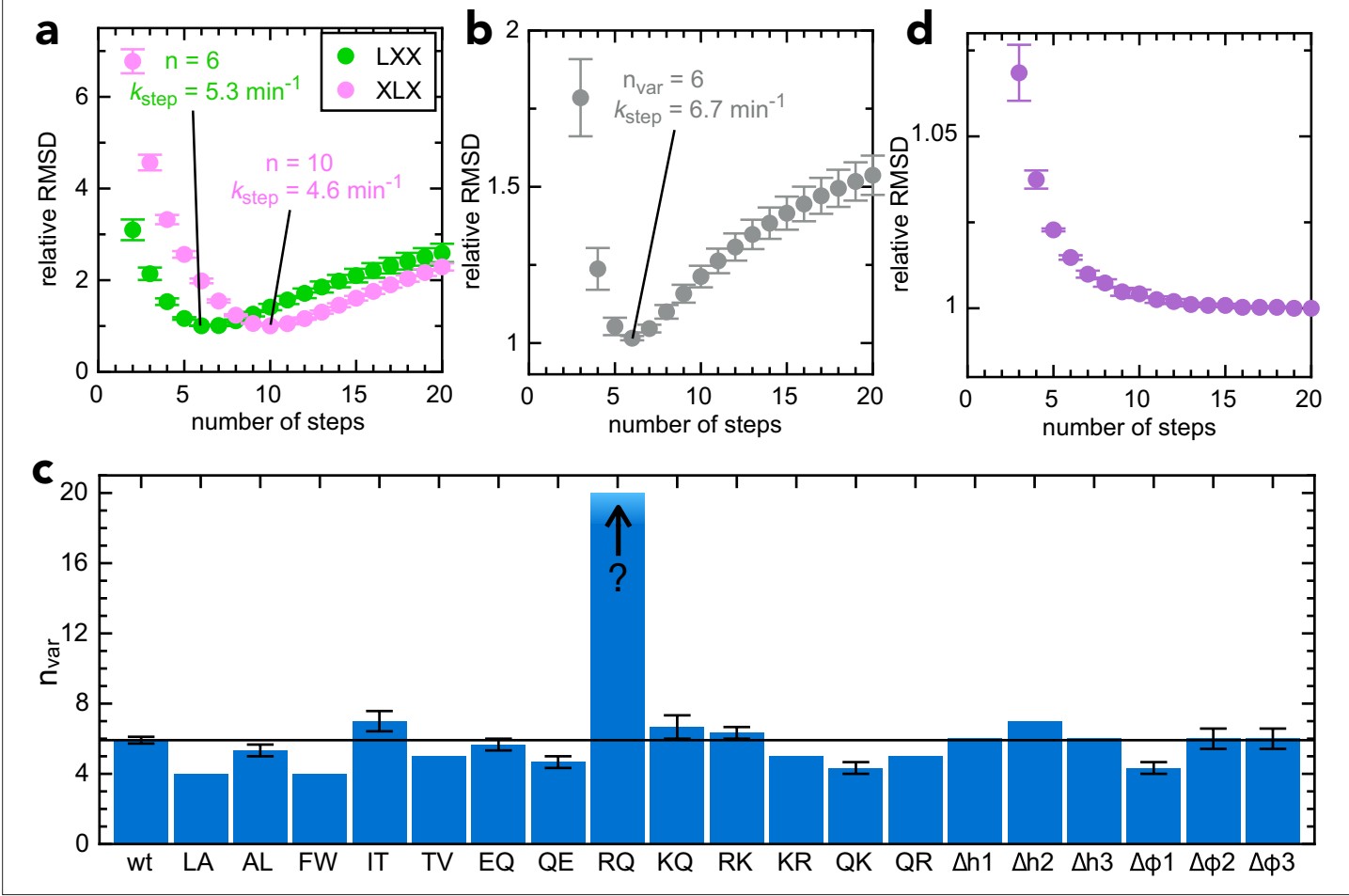

**Figure 3.** Determining step size for the variable mSpy regions. (**a**) RMSD (normalised to its lowest value) for fits to the experimental transport data to the original transport model (see *Figure 3—figure supplement 1a* and *Allen et al., 2020*), over a range of values for n (number of steps), using Berkeley Madonna. Error bars are the SEM from 4 (pSpy$_{LXX}$) or 12 (pSpy$_{XLX}$) replicates. Values for other fixed parameters are $k_{block}$ = 0.31 min$^{-1}$ (determined previously; *Allen et al., 2020*, $k_{on}$ = 0.96 µM$^{-1}$.min$^{-1}$ see *Figure 3—figure supplement 2a*) and $k_{off}$ = 0.085 min$^{-1}$ (see *Figure 3—figure supplement 2b*). Best fits are to 6 steps for pSpy$_{LXX}$ (green; $k_{step}$ = 5.31 ± 0.05 min$^{-1}$) and 10 steps for pSpy$_{XLX}$ (pink; $k_{step}$ = 4.57 ± 0.15 min$^{-1}$). (**b**) RMSD (normalised to its lowest value) for fits of experimental transport data for pSpy$_{XLX}$ to the model in *Figure 3—figure supplement 1b*, over a range of values for $n_{var}$. All parameters other than $k_{step,var}$, $k_{fail,var}$ and brightness are fixed to the same values as in panel (**a**). The best fit is to $n_{var}$ = 6, $k_{step,var}$ = 6.74 ± 0.24 min$^{-1}$. (**c**) Best fit $n_{var}$ for each of the pSpy$_{XLX}$ variants in *Figure 2d–j*, calculated as in panel (**b**), but with brightness adjusted for the values in *Figure 3—figure supplement 3* (see text for details), and $k_{block,var}$ allowed to float. Errors are the SEM from 12 (wt) or 3 (all others) replicates. (**d**) Normalised RMSD as a function of $n_{var}$ for pSpy$_{XLX}^{R \rightarrow Q}$ (all parameters as in panel **c**).

The online version of this article includes the following source data and figure supplement(s) for figure 3:

**Figure supplement 1.** Schematic representations of numerical models.

**Figure supplement 2.** Determination of $k_{on}$ and $k_{off}$ values for pSpy$_{XLX}$.

**Figure supplement 2—source data 1.** Raw data for *Figure 3—figure supplement 2a*.

**Figure supplement 2—source data 2.** Raw data for *Figure 3—figure supplement 2b*.

**Figure supplement 3.** Comparison of signal amplitudes for NanoLuc transport vs solution binding.

**Figure supplement 3—source data 1.** Raw data for *Figure 3—figure supplement 3*.

the best fit for pSpy$_{LXX}$ (n = 6, $k_{step}$ = 5.30 min$^{-1}$), while $k_{step}$ for the variant ($k_{step,var}$) was allowed to float and the best fit determined over a range of values for $n_{var}$. This gives a best fit of n = 6 for pSpy$_{XLX}$ ($k_{step}$ = 6.74 ± 0.24 min$^{-1}$; *Figure 3b*), in reasonable agreement with the fit to the simpler model (*Figure 3a*).

We next used this model to extract $n_{var}$ for each of the 19 pSpy$_{XLX}$ variants. The amplitudes of each variant (i.e. maximum signal, normalised to that of native pSpy$_{XLX}$ run in parallel) differ somewhat from sequence to sequence (*Figure 3—figure supplement 3*, grey bars) – due either to differences in the

signal produced by each NanoLuc ('brightness' in the Berkeley Madonna model) or in the probability that the sequences become trapped within the channel ($k_{block}$). To distinguish these possibilities, we measured the NanoLuc signal of each variant in solution, under saturating conditions (*Figure 3—figure supplement 3*, turquoise bars). The results suggest that in most but not all cases, the variance is down to small differences in NanoLuc brightness (compare grey and turquoise bars in *Figure 3—figure supplement 3*). To account for this in the fitting, we fixed brightness for each variant based on its measured value relative to native pSpy$_{XLX}$, then allowed $k_{block,var}$ to float, in addition to $k_{step,var}$ and $k_{fail,var}$.

The best fit number of steps for each variant is shown in *Figure 3c* (full fitting results are in *Supplementary file 3b*). In all but one case, n falls between 4 and 7 (i.e. very similar to the native sequence, n = 6). Strikingly, however, it is not possible to determine a number of steps for pSpy$_{XLX}^{R \rightarrow Q}$: the RMSD continues to go down then platueas as $n_{var}$ increases (at least up to 20; *Figure 3d*). We conclude that in the vast majority of cases transport time is dominated by a small number of slow steps; removal of arginines by mutagenesis eliminates these, revealing a large number of steps that are otherwise too fast to measure.

To summarise so far: arginines, which have a fixed positive charge, have profound effects on the process of transport; that is, in respect of both the rate and the average number of steps required to transport a given polypeptide across the membrane.

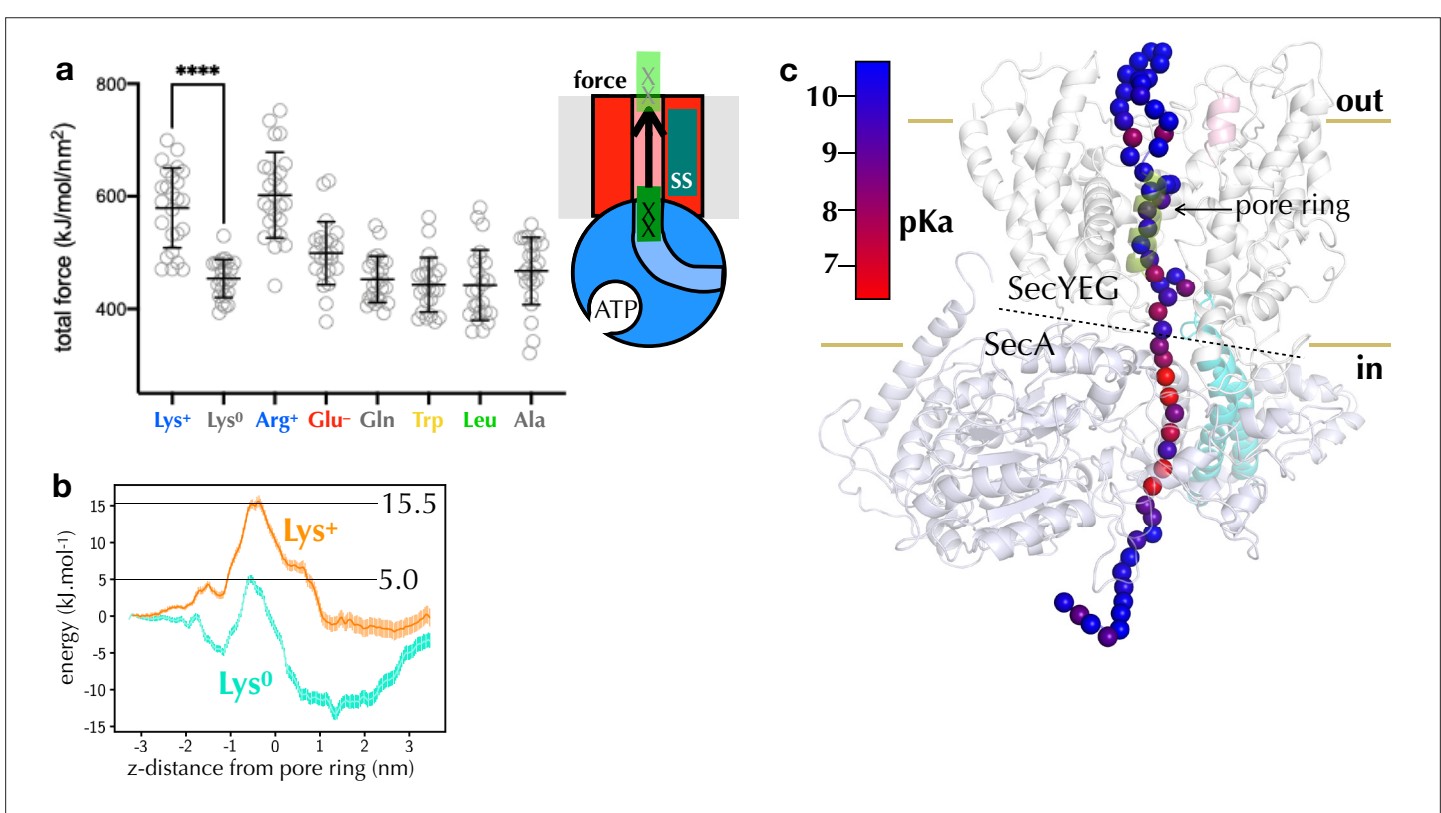

**Figure 4.** Computational analysis of pre-protein engaged in SecYEG. (**a**) Pulling forces for regions of polypeptide containing different residues of interest to pass the SecY pore, as determined using steered MD. Plotted are the integrated forces along the pull coordinated, with 24 repeats for each residue. The mean and standard deviations are shown. (**b**) Potential of mean force pathways for a short region of peptide with either protonated (Lys$^+$) or deprotonated (Lys$^0$) present passing through the SecY pore. These systems were built using the Martini force field. (**c**) p$K_a$ scanning data for lysine residues at different positions along the substrate. Alpha-carbon positions of the substrate are shown as spheres, and coloured according to their calculated p$K_a$.

The online version of this article includes the following source data for figure 4:

**Source data 1.** Raw data for panel a.

**Source data 2.** Raw data for panel b.

## Lysine is deprotonated to facilitate its passage through SecYEG

It has previously been shown that SecYEG is strongly selective against positively charged ions or residues, but more permeable to anions (*Dalal and Duong, 2009*; *Nouwen et al., 2009*; *Schiebel and Wickner, 1992*). However, the specific effect of arginine has not hitherto been described. To explore this further, we carried out constant velocity steered molecular dynamics (SMD), to explore how easy it is for different side chains to pass through the channel. Here, a 14-residue stretch of polypeptide, based on the PDB structure 5EUL (*Li et al., 2016*), containing a residue of interest is pulled through the SecY pore (see Materials and methods), and the total force experienced along this pull coordinate is recorded (*Figure 4a*). We found that the total force required to move positive charges across the membrane is significantly higher than any other residue type, but with no difference between protonated lysine and arginine (*Figure 4a*). The effect is specifically due to the positive charge: uncharged (deprotonated) lysine transports just as easily as other uncharged residues (e.g. $Lys^0$ vs Leu in *Figure 4a*). We quantified this effect using potential of mean force calculations with the coarse-grained Martini force field (*Marrink et al., 2007*; *Monticelli et al., 2008*), which predicts a difference in energy barrier of ca. 10 kJ mol$^{-1}$ between charged and uncharged lysine (*Figure 4b*).

The fact that lysine is transported more easily than arginine in vitro but not in silico would be explained if lysine loses its positive charge before traversing the channel. The p$K_a$ of lysine in solution is ~10, but it is readily deprotonated in hydrophobic environments (*Isom et al., 2011*), whereas the delocalised positive charge of arginine's guanidinium group (p$K_a$ ~13.8 [*Fitch et al., 2015*]) is much harder to remove (*Harms et al., 2011*). Additionally, the ΔpH component of the PMF would assist with the deprotonation of lysines at the entrance to SecY, and with their rapid reprotonation in the periplasm.

To explore this possibility, we carried out an in silico p$K_a$ analysis, in which a long stretch of native pre-protein substrate is threaded through the SecA-SecYEG channel (*Zimmer et al., 2008*), and relaxed over 1 μs of atomistic MD simulation (see Materials and methods). For multiple simulation snapshots, residues along the pre-protein were then mutated to lysine, relaxed with MD, and the p$K_a$ recorded using the propKa31 program (*Søndergaard et al., 2011*). The analysis reveals a clear region of p$K_a$ perturbation at the SecYEG-SecA interface (*Figure 4c*), in line with our predictions. Aside from their p$K_a$s, lysines and arginines are very similar in terms of their physical and chemical properties (circled in *Figure 2—figure supplement 2a*). The specific deprotonation of lysines therefore seems to be the only plausible explanation for the huge differential effect of arginine relative to lysine on transport time.

## Arginine transport exerts a selection pressure on secreted proteins

If transport of arginines is rate limiting for secretion in vivo, one might expect Sec substrates to experience an evolutionary selection pressure to eliminate arginines, where possible – most likely with lysine, the only other positively changed amino acid at neutral pH. To investigate this, we compared the Lys/Arg composition of the mature domain of all known *E. coli* Sec substrates with those that remain in the cytosol. As predicted, secreted proteins have a strong preference for lysine over arginine relative to those in the cytosol (*Figure 5a*). We also looked at pre-proteins that are exported, but by the Tat system – that is independently of Sec – and found their Lys/Arg ratio appears to fall somewhere between Sec substrates and unsecreted proteins (*Figure 5a*). This confirms that the Lys/Arg effect is at least in part a product of transport pathway, rather than the extra-cytosolic environment.

Because *E. coli* only uses Tat to export a very small number of proteins, we carried out the same analysis on a model organism with a large number of annotated Tat substrates (*Sinorhizobium meliloti*; *Pickering et al., 2012*; *Figure 5b*). The results both confirm the *E. coli* observations and show that they hold true across different classes of Gram-negative bacteria. The small difference in Lys/Arg ratio for Tat vs unsecreted proteins may be relevant to the mechanism of folded protein export, although it could equally reflect differences in the cytosolic vs periplasmic environment.

To explore whether this effect is related to ΔpH specifically, we carried out the same analysis on an alkalophilic organism with an inverted ΔpH (acid$_{in}$/alkaline$_{out}$), *Bacillus halodurans* (*Takami et al., 2000*). Consistent with ΔpH being involved in transport, lysine is no longer favoured over arginine for secreted proteins – indeed, the reverse appears to be true (*Figure 5c*). Meanwhile, the related bacterium *Bacillus subtilis*, which grows in neutral environments (alkaline$_{in}$/acid$_{out}$), shows the expected

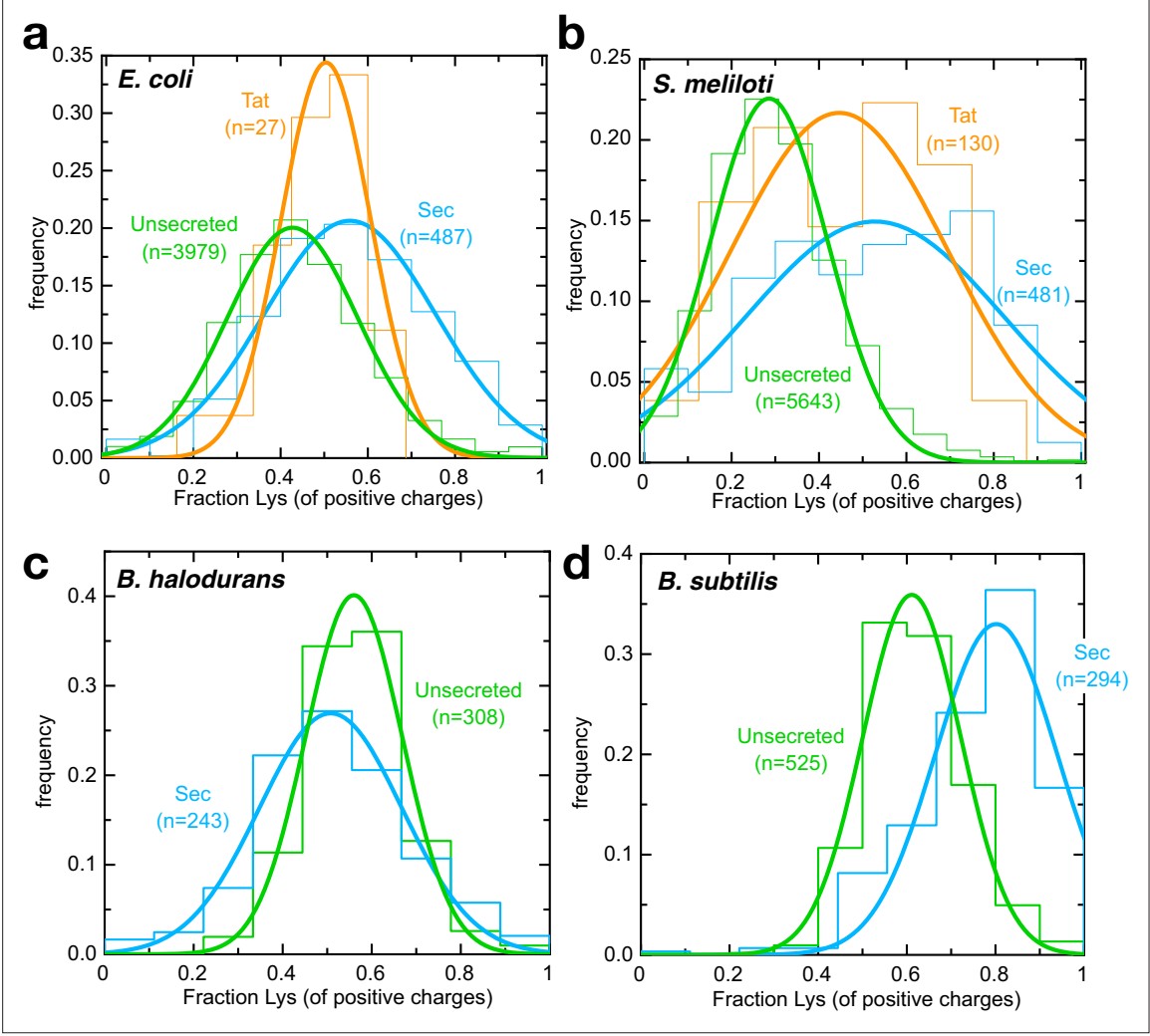

**Figure 5.** Arginines are selected against for secretion by neutrophiles. (**a–d**) Histograms (fine lines) showing fraction of positive residues that are lysine in cytosolic proteins (green), Sec substrates (blue) and Tat substrates (orange; panels **a,b** only), for (**a**) *E. coli*, (**b**) *S. meliloti*, (**c**) *B. halodurans* and (**d**) *B. subtilis*. Best fit single Gaussians are also shown (thick lines) for clarity. All differences are statistically significant except for *E. coli* Tat substrates; p-values are shown in **Supplementary file 3c**.

The online version of this article includes the following source data for figure 5:

**Source data 1.** Binned Lys/Arg ratios for **Figure 5**.

increase in preference for lysine for secreted substrates (**Figure 5d**). The fact that this trend holds true in three very distantly related bacteria suggests it could be a general feature of secreted proteins.

## Proton motive force speeds up transport primarily through Δψ

The question of how the PMF stimulates SecA-mediated pre-protein transport has been open for decades (**Brundage et al., 1990**): while it is intuitive to imagine negatively charged residues crossing the membrane electrophoretically with the aid of $\Delta\phi$, the same effect would equally prevent transport of positive charges. Indeed, a strong electrophoretic pulling force has been observed for negatively charged residues at the SecY channel entrance during co-translational membrane protein insertion, but no equivalent slowing of positively charged residues (**Ismail et al., 2015**). Deprotonation of lysines at the cytosolic face of SecYEG could neatly circumvent this problem, by imbuing essentially all pre-proteins with a net negative charge as they cross the membrane. To investigate this possibility, we therefore set out to measure the effect of PMF on transport.

To generate a continuous and stable PMF, we switched from PLs to IMVs purified from cells (over-) producing SecYEG and 11S (see *Pereira et al., 2019*). IMVs derived from normally functioning *E. coli* strains contain $F_1F_o$-ATP synthase, which works in reverse to produce a PMF upon addition of exogenous ATP. IMVs differ from PLs in that they contain other, native inner membrane proteins, albeit at much lower stoichiometry relative to SecYEG compared to native membranes. We started by comparing transport of pSpy-pep86 into PLs (*Figure 6a*, pink) with transport into IMVs derived from a cell line lacking $F_1F_o$-ATP synthase, and therefore unable to generate PMF from ATP (HB1; green in *Figure 6a*). The results show that lag – which corresponds to transport time (see *Figure 2b* and *Allen et al., 2020*) – is identical for both, but IMV transport reaches completion faster, meaning a lower probability that transport stalls or fails. Presumably, this enhanced transport processivity is caused by differences between IMVs and PLs – either a specific effect of auxilliary Sec components, or non-specific differences in the membrane environment.

In order to compare transport with and without PMF under otherwise identical conditions, we used IMVs with functional $F_1F_o$-ATP synthase (BL21) and either added (–PMF) or omitted (+PMF) two iono-phores: valinomycin, a potassium ionophore that specifically depletes $\Delta\psi$ when $K^+$ is present in the buffer; and nigericin, an electroneutral $K^+/H^+$ antiporter that dissipates the $\Delta$pH only under the same conditions. Together, these should eliminate PMF entirely; and indeed, as expected, when both iono-phores are present transport into BL21 IMVs give a similar lag to PLs or HB1 IMVs (grey in *Figure 6a*, see also inset). In the absence of ionophores (i.e. +PMF), transport (lag) is about about twice as fast as in their presence (orange vs grey in *Figure 6b*). Surprisingly, however, the total amplitude is slightly higher with the ionophores. As import is largely single turnover under the experimental conditions used (one pre-protein per SecYEG), an increase in amplitude suggests either that pre-proteins are less likely to become irrevocably stuck in the channel during transport, or that more of the SecYEG sites are active. Alternatively, the lowered pH inside the PL lumen in the presence of PMF might cause a reduction of NanoLuc signal – either by affecting NanoLuc directly or reducing the accessibility of pep86 – that is reversed by nigericin.

To separate the effects of $\Delta\psi$ and $\Delta$pH, we next measured import with each of the two ionophores individually. Valinomycin alone (no $\Delta\psi$, $\Delta$pH increases to compensate) slows transport and produces a small reduction in amplitude (blue in *Figure 6b*), while nigericin (no $\Delta$pH, increased $\Delta\psi$) appears both to speed up transport and increase the maximum amplitude (green in *Figure 6b*). These effects are not caused by non-specific effects of the ionophores, as they are not observed for transport into HB1 IMVs (*Figure 6—figure supplement 1a*). The faster import with nigericin most likely arises from an increase in $\Delta\psi$ caused by the dissipation of $\Delta$pH; but this cannot explain the increase in amplitude, which is maintained even in the presence of valinomycin. A reasonable hypothesis is that nigericin is alleviating an effect on signal from the lowered pH in the energised IMV lumen.

While the effects of PMF and the individual ionophores are consistent and reproducible, they are relatively small – certainly not sufficient to bridge the difference in rate between ATP-only driven transport in vitro and the estimated two orders of magnitude faster transport rate in vivo (*Allen et al., 2020*; *Cranford-Smith and Huber, 2018*). We therefore measured the magnitude of PMF and its individual components in our purified IMVs, and compared them to the cells from which they are derived (*Figure 6c*; see Materials and methods for details). The results suggest that the $\Delta\psi$ produced by the reverse action of ATP synthase in purified IMVs is comparable to that of intact, respiring *E. coli*, and that overexpression of SecYEG also makes little difference (*Figure 6c*). However, the absolute measured value of $\Delta\psi$ is much lower than is generally reported for *E. coli* (~150 mV; *McMillan et al., 2007*), and it is also possible that protein transport itself consumes PMF in vitro, as we have observed for mitochondrial protein import (*Ford et al., 2021*). Therefore, we cannot conclusively determine the extent to which PMF contributes to pre-protein transport in the absence of auxilliary factors.

## Stimulation of transport by PMF is pH-dependent

The IMV lumen volume is only a tiny fraction of the total reaction volume in the transport assay (< 1/5000), so $\Delta$pH presumably manifests as pH decrease inside the vesicle with negligible effect on the bulk pH outside. This contrasts with the situation in a living cell, where $\Delta$pH primarily affects the pH on the cytosolic side of the membrane. Yet if deprotonation of lysines is part of the mechanism of PMF stimulation, then it is the pH experienced by the pre-protein before it is transported that matters – perhaps explaining why nigericin does not slow down transport in vitro.

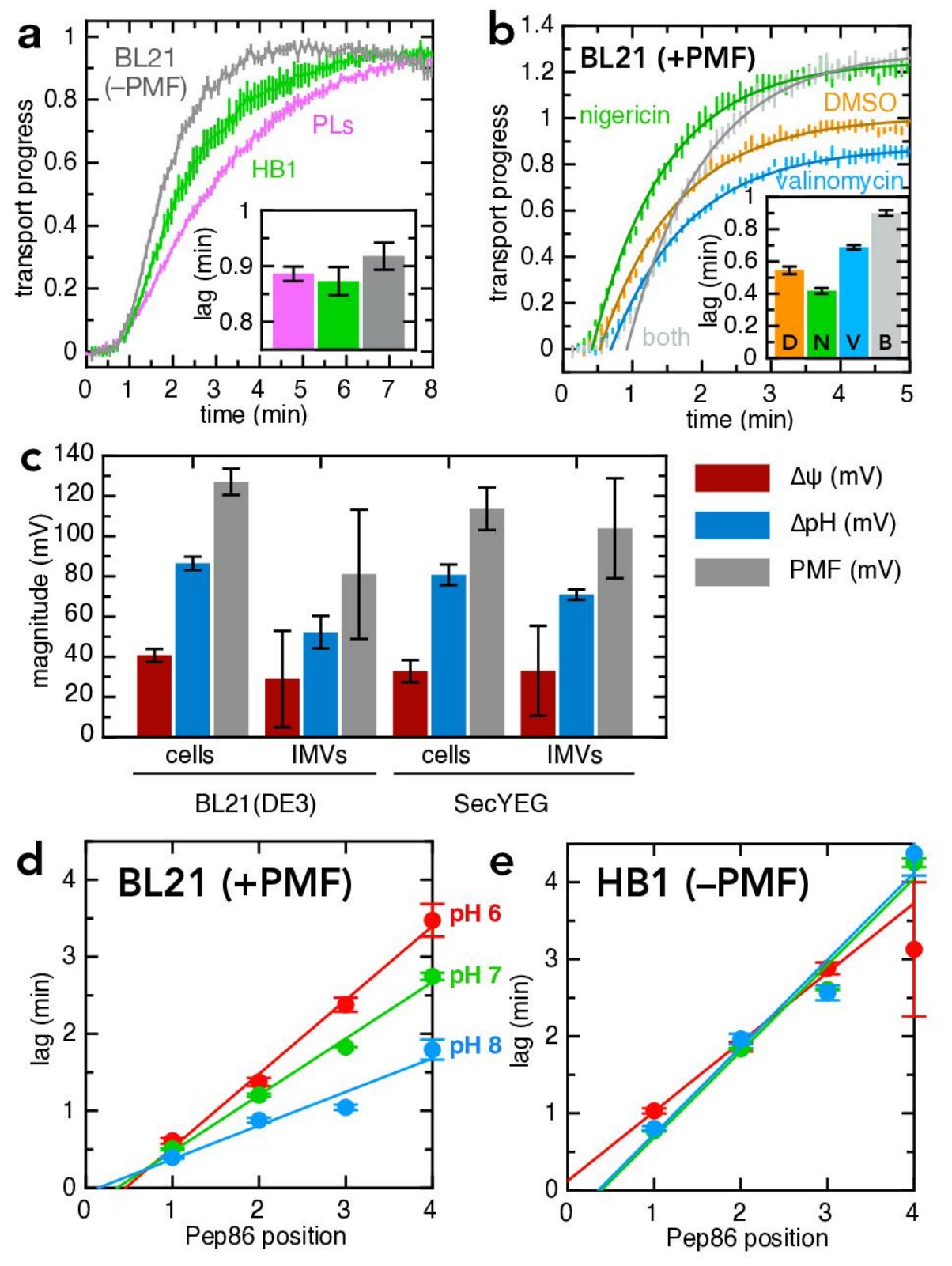

**Figure 6.** PMF stimulates transport primarily via Δφ. (**a**) Transport of pSpy-pep86 into PLs (pink), HB1 IMVs (lacking ATP synthase; green) and BL21 IMVs in the presence of valinomycin and nigericin (grey). Data points and their heights represent the average and SEM from three (PLs), six (HB1) and six (BL21) experimental replicates. Best fit lags (zoomed in for clarity) are shown as an inset; none of the differences in lag are statistically significant as determined by a two-tailed t-test (p = 0.69, 0.31 and 0.20, for PLs-HB1, PLs-BL21, and HB1-BL21, respectively). (**b**) Transport of pSpy-pep86 into BL21

*Figure 6 continued on next page*

*Figure 6 continued*

IMVs in the presence of DMSO (orange), valinomycin (blue), nigericin (green), or both ionophores (grey). Data points and their heights represent the average and SEM from six replicates, and lines show the best fit to the single exponential + lag model. Best fit lags are shown inset, and all ionophores have a statistically significant effect on lag as determined by a two-tailed t-test (p = 0.0015, 0.00041 and $3.6 \times 10^{-7}$ for nigericin, valinomycin and both, respectively). (**c**) Comparison of the proton-motive force (PMF, grey) and its two components ($\Delta\phi$, red and $\Delta$pH, blue), generated by whole cells and inverted membrane vesicles (IMVs), in BL21(DE3) cells both normal and overexpressing SecYEG. Error bars indicate the standard deviation from three biological replicates (for cells) or three technical replicates (for IMVs). Underlying data and statistics shown in *Supplementary file 3d*; note that only total PMF in pBAD cells vs IMVs is statistically different (p = 0.0084). (**d**) Lags for import of the pSpy$_{4x}$ series as a function of active pep86 position, into BL21 IMVs at pH 6 (red), 7 (green), and 8 (blue). Data and error bars are the average and SEM from three replicates. (**e**) As in panel (d), but with HB1 IMVs.

The online version of this article includes the following source data and figure supplement(s) for figure 6:

**Source data 1.** Raw data for *Figure 6a–b*, and *Figure 6—figure supplement 1*.

**Source data 2.** Raw data for *Figure 6c*.

**Source data 3.** Raw data for *Figure 6d*.

**Source data 4.** Raw data for *Figure 6e*.

**Figure supplement 1.** Exactly as in *Figure 6b*, but with HB1 IMVs instead of BL21 IMVs.

**Figure supplement 2.** Additional transport data for the pSpy$_{4x}$ series.

**Figure supplement 2—source data 1.** Raw data for *Figure 6—figure supplement 2a-c*.

To test the effect of topologically cytoplasmic pH, we prepared IMVs at three different pHs (6.0, 7.0, and 8.0) and measured import in the same buffer. In the absence of PMF, both the internal and external pH should be the same, and equal to the starting value; generating a PMF will acidify the vesicle lumen but leave the outside pH essentially unaffected. To test the effect of this on transport rate and PMF stimulation we used the pSpy$_{4x}$ series described previously (*Allen et al., 2020*). This comprises four proteins identical except for the number of copies of mSpy proteins that must pass through SecY before active pep86 becomes available to bind 11S; a plot of lag against active pep86 position therefore gives a straight line with a slope equal to the transport rate in Spy.min$^{-1}$ (*Allen et al., 2020*).

Just as observed for native pSpy (above), transport is considerably faster (~2 fold) with PMF (BL21 IMVs) than without (HB1 IMVs) at pH 8.0 (*Figure 6d* vs *Figure 6e*, blue data). However, as pH decreases, this difference vanishes: at pH 6.0, there is very little PMF effect on rate (*Figure 6d–e*). The same effect is observed when using ionophores, with the stimulatory effect of nigericin on amplitude also amplified at high pH (*Figure 6—figure supplement 2a-c*). While this does not prove that it is specifically the pre-protein that is deprotonated, it is evidence that the mechanism of transport stimulation by $\Delta\phi$ is dependent on cytosolic pH.

It should also be noted that the overall import signal is much lower at low pH (*Figure 6—figure supplement 2d-e*). This amplitude reflects the number of functional SecYEG import sites, the probability of transport blocking and the activity of the formed NanoLuc – so a number of things could be causing this reduction. These include faster deterioration of SecYEG during the IMV preparation or lower NanoLuc signal at low pH. Once total amplitude is accounted for, however, the rate at which pSpy$_{4x}$ becomes irreversibly trapped in the channel appears to be largely unaffected by pH (*Figure 6—figure supplement 2f-g*), ruling this out as a cause.

## PMF stimulation of transport is not measurably dependent on pre-protein charge

If the PMF primarily is acting on charged residues – promoting transport of negative changes and inhibiting diffusion of arginines – we would expect the magnitude of PMF stimulation to be dependent on the number of charged residues. We therefore measured transport of each of the pSpy$_{XLX}$ variants into BL21 IMVs, both in the presence and absence of valinomycin and nigericin, and calculated the stimulatory effect of PMF on transport time for each (see Materials and methods). The results are shown in *Figure 7a–f*, with 0% meaning no effect of PMF and 100% a halving of lag in the presence of PMF. As expected, transport is faster in the presence of PMF for all variants. Unexpectedly, however, we see little indication that PMF is acting on charged residues. Indeed, all variants where the number

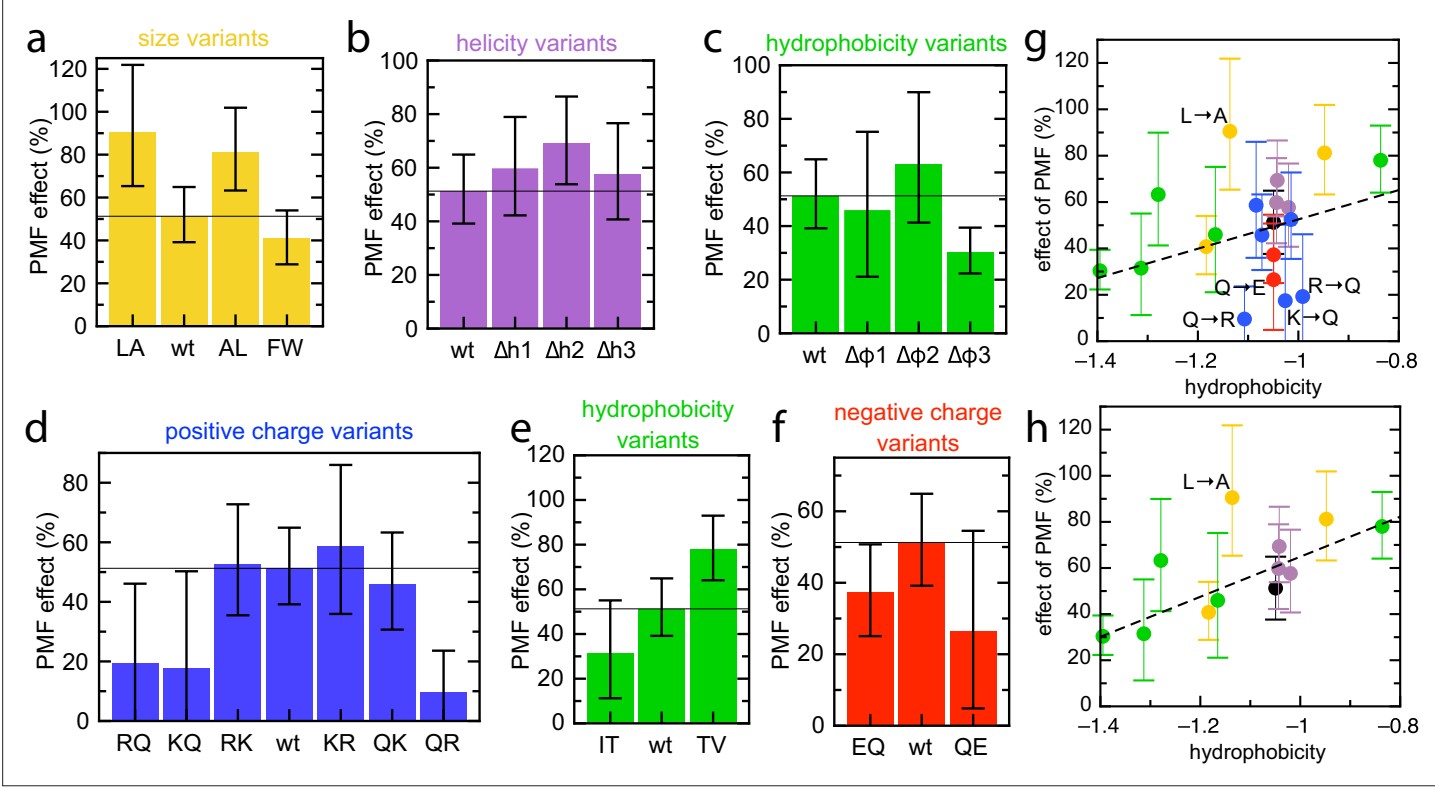

**Figure 7.** Effect of PMF on transport of the pSpy_XLX variants. (**a–f**) Stimulatory effect of PMF on transport of the variable region of all the pSpy_XLX variants. These were calculated from the difference in lag for import into BL21 IMVs in the presence and absence of valinomycin and nigericin (see Materials and methods for details and **Figure 7—figure supplement 1** for transport times), where 0% is no difference and 100% is a halving of lag. Error bars are calculated by calculating minimum and maximum values from the SEMs from five replicates of each. (**g,h**) PMF effects (coloured as in panels **a–f**) as a function of average hydrophobicity score for the amino acids in each variable mSpy, calculated using the values in **Kyte and Doolittle, 1982**. (**g**) Entire data set, with best fit line (r = 0.271). (**h**) Data excluding the charge variants (i.e. the blue and red values; r = 0.681). Outliers are marked directly on the plots.

The online version of this article includes the following source data and figure supplement(s) for figure 7:

**Source data 1.** Raw data for **Figure 7** without ionophores.

**Source data 2.** Raw data for **Figure 7** with ionophores.

**Figure supplement 1.** Complete lag data for the pSpy variants in different vesicle types.

of charges is altered show reduced PMF effect relative to native mSpy – even Q→E, which one might reasonably expect to be assisted by $\Delta\phi$ regardless of other factors.

The only amino acid property that appears to correlate with magnitude of the PMF effect is hydrophobicity, in that more hydrophobic variants exhibit higher stimulation by PMF. A scatter plot of PMF effect vs hydrophobicity shows this more clearly – a straight line through all the points shows a weak correlation (**Figure 7g**; r = 0.271), which becomes stronger if the charge variants, which mostly show reduced PMF effect, are omitted (**Figure 7h**; r = 0.681). While the large error bars (caused by dividing two experimentally determined numbers) preclude confident assignment of PMF effect to hydrophobic residues, it is the only general amino acid property that has any noticeable effect on this parameter.

Although it is certainly possible that PMF is acting on hydrophobic residues, for example by promoting conformational changes in SecY, the correlation is weak and it would be surprising to us if this were a more significant factor than electrophoresis – particularly given that charged residues in the loops of membrane proteins do experience $\Delta\phi$ from the opposite side of the membrane (**Ismail et al., 2015**). A more plausible explanation for this result is that the PMF effect in the IMV assay is confounded by some other factor. For example, there might be differences in folding behaviour that affect how the pre-protein interacts with the Sec system; a threshold effect of PMF that is not reached in our in vitro system (see **Figure 6c**); or some contribution by other, unidentified components of IMVs.

The exact relationship between PMF, the translocation machinery and pre-protein sequence therefore remains an open question.

## Discussion

The landmark solution of the first atomic structure of SecYEG in complex with SecA, over a decade ago (*Zimmer et al., 2008*), inspired multiple hypotheses for the underlying mechanism for the transport of pre-proteins across membranes. However, the difficulties obtaining quantitative data on transport itself have, until recently, prevented these from being properly tested. Here, we have applied the newly developed, high time-resolution NanoLuc transport assay (*Pereira et al., 2019*) to a set of model pre-proteins, designed to capture the rate of transport for amino acid sequences with different properties. Our results reveal a surprisingly simple set of rules for how fast pre-protein is transported through SecYEG: arginine is by far the slowest amino acid to transport, accounting for over 2/3 of the total transport time of the model pre-protein pSpy. Other residues that make a difference are lysines, and bulky residues such as tryptophan, with diffusion of bulky and positive patches probably accounting for the individual steps observed in transport (*Figure 1*, purple balls).

In our experiments, transport of arginines is slow and rate-limiting. In contrast, lysines behave much more like neutral residues, suggesting that they are at least partially deprotonated before passage across the membrane, a proposition supported by computational analysis of the SecYEG-A structure bound to various model pre-cursors. This deprotonation is presumably facilitated by SecA, with the energy required coming either from ΔpH – protons are more readily removed at the higher pH on the cytosolic side of the membrane – and/or the hydrolytic cycle of ATP, with nucleotide state-induced

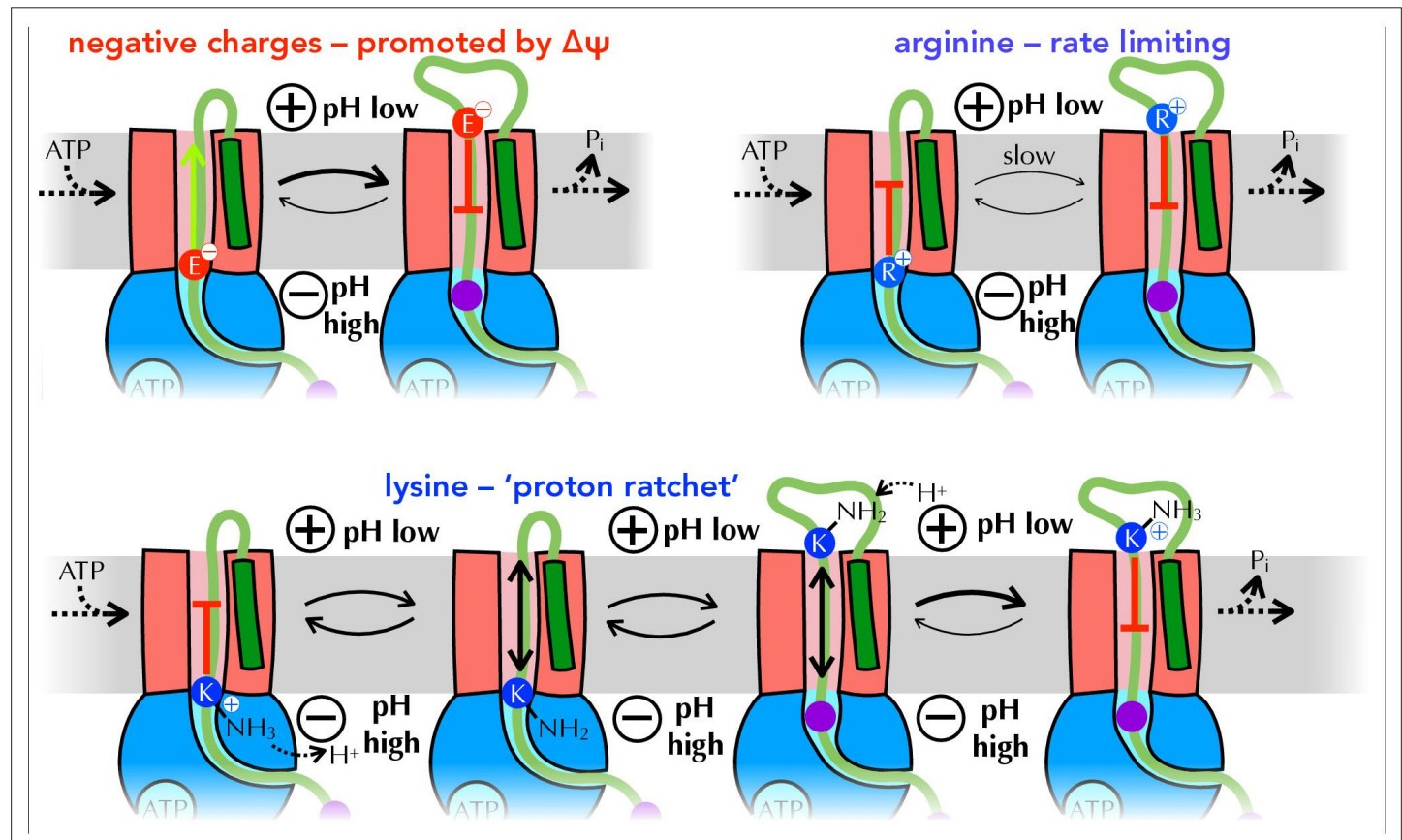

**Figure 8.** Summary of proposed PMF and ratcheting effects of pre-protein transport. Closeup of step *iii* from *Figure 1*, where a negative amino acid (red, top left), arginine (blue, top right) or lysine (blue, bottom) is present at the entrance to SecY. Δ$\psi$ is expected to promote the forward diffusion of negative charges, and inhibit their reverse movement. Movement of arginine is unfavourable in either direction, with export especially unfavourable when Δ$\psi$ is present; hence transport of arginine is rate-limiting for the entire process. Lysine is deprotonated at the cytosolic face of SecY (by ΔpH and/ or ATP hydrolysis by SecA), allowing its free diffusion across the membrane, and preventing backsliding once it is reprotonated on the other side.

conformational changes affecting the p$K_a$ of lysines in the channel. The fact that ΔpH does not appear to have a major effect on transport rate would suggest that the latter is more important, although we interpret these results with caution: the way ΔpH manifests in the assay (lower pH at the channel exit) is not necessarily exactly the same as in vivo (primarily elevated pH at the channel entrance), and our IMV preparations have a low total PMF compared to literature values for *E. coli* cells.

In addition to allowing easy passage of lysine through the SecY channel, which is strongly selective against cations, deprotonation could stimulate transport in two other ways. Firstly, it gives all translocating pre-proteins a net negative charge: as illustrated in *Figure 8*, diffusion of negative charges (aspartic and glutamic acid) is likely to be promoted by $\Delta\phi$ (*Ismail et al., 2015*), while only arginines (which are under-represented in secreted proteins, see *Figure 5*) inhibit transport. Additionally, once deprotonated lysines emerge from SecY into the lower pH environment of the periplasm and absent the p$K_a$ perturbing properties of SecA, they will swiftly be reprotonated and thus unable to diffuse back. This 'proton ratchet' effect will prevent backsliding of transported lysines, and thereby contribute to the forward progression of translocation (*Figure 8*).

The computational results above were derived from peptides with no other charged residues nearby. Presumably, however, sequence context will have an effect in real life; for example a positive charge adjacent to a negative charge might be harder to deprotonate but easier to transport through the channel, as the two partially cancel each other out. Conversely, a long stretch of consecutive lysines might overwhelm the ability of the Sec machinery to strip and dissipate protons. Poly-lysine is not generally a feature of secreted proteins, but it has previously been used as evidence that positively charged residues are hard to transport (*Liang et al., 2012*; *Nouwen et al., 2009*); possibly this strong effect is conferred by the presence of multiple consecutive lysines, as opposed to ones spread evenly through the sequence. The importance of detailed sequence context, along with the compromised PMF in purified IMVs, might also explain the apparently contradictory results we obtain for PMF stimulation of charged pSpy$_{XLX}$ variants.

While arginine has by far the most striking effect of any amino acid, other substitutions also impact the rate of transport. Large amino acids are transported more slowly than smaller ones, most likely because there is more resistance to them diffusing though SecY. Knocking out stretches of α-helix also slows transport, consistent with the Sec system acting on these specifically (*Corey et al., 2019*). The effect of hydrophobicity, however, is less clear cut: while more hydrophobic sequences are generally transported more slowly, the correlation is not absolute, and knocking out hydrophobic patches has little if any effect on transport. Furthermore, the assay does not allow us to distinguish between general mechanistic effects – for example hydrophobic sequences passing through the channel more slowly – and sequence-specific effects, such as specific hydrogen bonds inhibiting passage of a single part of the pre-protein. Nor are we able to measure the intrinsic structure of the pre-protein before it is transported (*Best, 2020*), although this is likely to be affected by sequence and in turn affect the rate of diffusion through the channel. Resolving these issues will most likely require studying transport of very short protein sequences at the single molecule level.

In addition to the direct mechanistic insights that can be discerned from the above results, they also have more general implications for the nature of protein secretion. It has previously been shown that a lower propensity for folding and the presence of hydrophobic patches are hallmarks of bacterial secreted proteins (*Chatzi et al., 2017*); to this we can now add a reduction in the number of arginines. Intriguingly, it also seems that any alterations to the charged residues in Spy – either adding or removing positive or negative charges – reduces the magnitude of PMF stimulation. This perhaps indicates that the sequence of Spy is already well optimised for secretion in terms of charge distribution. These observations all imply that 'secretability' is an important evolutionary constraint on proteins that localise outside the cytosol, particularly very abundant proteins such as Spy (which can account for ~25% of periplasmic upon induction *Quan et al., 2011*). This will have important ramifications for the rational design of secretion-competent proteins for biotechnology or synthetic biology applications. Our results further suggest that replacing arginines with lysines will be a simple but effective way to achieve higher secretability.

Above all, the identification of steps in the translocation process through SecYEG is important because their properties enable us to distinguish between diffusional (described above) and power-stroke (*Catipovic et al., 2019*; *Catipovic and Rapoport, 2020*) mechanisms of translocation. The steps of the translocation reaction we identify could in theory correspond either to average ratchet

lengths during diffusion, or to individual piston motions of a power-stroke. But in the former case the steps would vary with the properties of the translocating pre-protein, as indeed they do; steps in a power stroke, meanwhile, would depend on the geometry of SecA and the conformational changes associated with ATP turnover, and thus be invariant. Ratcheting has the further benefit of allowing other factors to contribute to pre-protein transport, including the PMF; pre-protein folding on the outside over the inside (*Corey et al., 2019*); and the action of auxiliary Sec components such as SecDF (*Tsukazaki, 2018*) and periplasmic chaperones (*Fürst et al., 2018*; *Knyazev et al., 2018*).

The observation that SecA is required to allow positively charged lysines through the SecY channel might plausibly also have implications for topogenesis of membrane proteins. The 'positive-inside rule' (*Cymer et al., 2015*; *von Heijne, 1989*) – that loops with more positive character are retained in the cytosol – may in part reflect slower kinetics of transporting such loops across the membrane.

## Materials and methods

### Translocation substrate production

To produce the pSpy$_{XLX}$ variants, genes for each variant pSpy were first either synthesised (GeneArt, Thermo Fisher Scientific; pSpy$_{R\to Q}$, pSpy$_{K\to Q}$, pSpy$_{R\to K}$, pSpy$_{K\to R}$, pSpy$_{Q\to K}$, pSpy$_{Q\to R}$, pSpy$_{E\to Q}$, pSpy$_{Q\to E}$, pSpy$_{A\to L}$, pSpy$_{L\to A}$, pSpy$_{I\to T}$, pSpy$_{T\to V}$) or the mutations introduced by site-directed mutagenesis (Quik-Change, Agilent; pSpy$_{\Delta h1}$, pSpy$_{\Delta h2}$, pSpy$_{\Delta h3}$, pSpy$_{\Delta\varphi1}$, pSpy$_{\Delta\varphi2}$, pSpy$_{\Delta\varphi3}$, pSpy$_{F\to W}$). A gene for pSpy$_{LX}$ was created by removing the dark peptide from pSpy$_{LD}$ (in pBAD/*myc*-His C; *Allen et al., 2020*) using QuikChange. Each variant mSpy was then cloned between the first mSpy and the pep86 sequences using site directed ligase independent mutagenesis (*Chiu et al., 2004*). For pSpy$_{LXX}$, the wild type mSpy gene was instead cloned after the pep86 sequence. A complete list of new protein sequences is shown in *Supplementary file 1*.

All translocation substrates (including the pSpy$_{4x}$ variants; *Allen et al., 2020*) were expressed as for native pSpy (*Pereira et al., 2019*), then purified using a new, streamlined protocol. Cell pellets were resuspended 1:7 (v/v) in 7 M guanidine.HCl on ice, incubated for 30 min, then centrifuged at 100,000 g for 30 min. The supernatents were next bound to Ni$^{+2}$ affinity resin (~10 x bead volume) by rotating gently for 30 min at 4 °C, followed by decanting into empty 10 ml gravity flow columns and washing with ≥5 bed volumes tris/urea buffer (20 mM Tris-HCl pH 8.0, 6 M urea) supplemented with 30 mM imidazole. Protein was eluted using tris/urea buffer supplemented with 330 mM imizadole, then passed over a Q-sepharose column to remove nucleic acid contamination. Finally, the purified proteins were concentrated and imidazole removed by spin concentration.

### Production of other translocation reagents

Inverted membrane vesicles (IMVs) overexpressing SecYEG and membrane-tethered 11S (~11S) were produced from previously described cells strains (*Pereira et al., 2019*). Cells were grown at 37 °C to mid-log phase (OD$_{600}$ ~0.6) in 2xYT supplemented with 100 mg.ml$^{-1}$ amplicillin and 50 mg.ml$^{-1}$ kanamycin, then SecYEG expression was induced with 0.1% (w/v) arabinose. After 2 hr, the cells were cooled to 20 °C then ~11S expression induced overnight (~16 hr) with 1 mM IPTG before harvesting. For purification, a standard protocol was followed (*Corey et al., 2018*), but using as buffer either M$_6$KM (20 mM MES-KOH pH 6.0, 50 mM KCl, 2 mM MgCl$_2$), H$_7$KM (20 mM HEPES-KOH pH 6.0, 50 mM KCl, 2 mM MgCl$_2$) or M$_8$KM (20 mM MOPS-KOH pH 8.0, 50 mM KCl, 2 mM MgCl$_2$), depending on the desired final pH.

All other transport reaction components were produced as described previously (*Pereira et al., 2019*). For PLs, this entailed solubilising dried *E. coli* polar lipid (Avanti) in TKM (20 mM Tris-HCl pH 8.0, 50 mM KCl, 2 mM MgCl$_2$) with 1% DDM to 4 mg.ml$^{-1}$, then mixing with purified SecYEG at 3.3 μM and 11S at 20 μM. The mix was then extruded to 400 nm and dialysed (12–14 kDa MWCO) overnight in the cold room (4°C–8°C) against TKM, with buffer exchanges after ~2 hr and ~5 hr. PLs were harvested by centrifugation at 100,000 g for 20 min, then washed three times by resuspending in TKM and recentrifuging. The final PL pellets were resuspended to 1:10 weight/volume with TKM.

### Transport assays

Transport assays were performed and analysed exactly as in *Allen et al., 2020*, with 2 μM final concentration of pre-protein (unless otherwise stated) to ensure saturation. All raw NanoLuc experimental

traces are included as Source Data. For IMV experiments, IMVs and the appropriate pH buffer ($M_6KM$, $H_7KM$ or $M_8KM$; pH 8 used unless otherwise stated) were substituted for PLs and TKM, but all other reaction conditions were kept identical. Where ionophores were used, they were added from a 100 x stock in DMSO to final concentrations of 1 µM (valinomycin) or 2 µM (nigericin), and the corresponding amount of DMSO was added to comparison experiments.

For Berkeley Madonna analysis of the $pSpy_{XLX}$ variants, the model in *Allen et al., 2020* was modified to allow different translocation parameters for the native and variant sequences. A visual representation of this new model is shown in *Figure 3—figure supplement 1b*; the complete model is shown in *Supplementary file 2*.

To calculate PMF effect, the import lag for $pSpy_{LXX}$ was subtracted from the lag for each $pSpy_{XLX}$ variant, both in the absence (+PMF) and presence (–PMF) of nigericin and valinomycin, to give transport time just for the variable region. We then calculated PMF effect (in %) as:

$$PMF\,effect = 100 \times \left( \frac{lag_{-PMF}}{lag_{+PMF}} - 1 \right)$$

Errors in lag were take as the SEM of lag from four technical replicates, where +PMF and –PMF were identical other than the addition of ionophores, and error bars for PMF effect were generated by calculating upper and lower bounds from these errors.

## Sequence analysis

Protein properties were calculated over a 9 residue window using ProtScale (https://web.expasy.org/protscale/), using values for hydrophobicity from *Kyte and Doolittle, 1982*, helical propensity from *Deléage and Roux, 1987* and bulkiness from *Zimmerman et al., 1968*. The same scales were used to plot *Figure 2—figure supplement 2a*.

## Arginine/Lysine ratio determination

The complete proteomes of *E. coli* (strain K12) and *S. meliloti* (strain 1021) were downloaded from UniProt (*The UniProt Consortium, 2017*) on 1st May 2018 and sorted according whether they are secreted by Sec or Tat system, or unsecreted. For *E. coli*, which is well annotated by UniProt, this was done using information in the 'Signal Peptide' column. For *S. meliloti*, secreted substrates were selected according to UniProt 'Signal Peptide', then classified as Tat if validated as such in *Pickering et al., 2012*, or Sec otherwise. *B. halodurans* and *B. subtilis* sequences were downloaded on 13th March 2019 and analysed as for *E. coli*, omitting signal peptides flagged as 'Tat-type'. For secretion substrates, the residues corresponding to the SS were removed prior to analysis. The proportion of positive residues that are lysine (Lys/(Lys + Arg)) was calculated for each protein, then plotted as a histogram for each data set (with sample size n): the number of bins was determined by Sturge's rule (number of bins = 1 + 3.322 * log(n)), and frequency calculated by dividing the population of that bin by n. Fits to single Gaussian curves were performed using Pro Fit (Quansoft).

## Molecular dynamics of substrate moving through SecY

Systems were built using the coordinates of SecYE and a 14 residue stretch of preprotein (residues 778–791) from PDB 5EUL (*Li et al., 2016*). The pre-protein was changed into a model peptide, with the sequence AGSGSGSGSGGXGA, where X is the residue of interest (K, R, E, Q, W, L or A). The protein coordinates were built into a POPE:POPG membrane using CHARMM-GUI (*Jo et al., 2007*; *Lee et al., 2016*). Proteins were described using the CHARMM36m force field (*Best et al., 2012*), and waters were TIP3P, with $K^+$ and $Cl^-$ ions added to 0.15 M. The protein side chains were set to their default protonation states, as predicted using propKa3 (*Søndergaard et al., 2011*), apart from the $Lys^0$ residue where included. Systems were energy minimized using the steepest descents method, and subsequently equilibrated with 1000 $kJ.mol^{-1}.nm^{-2}$ positional restraints on protein backbone atoms for 5 ns, and then relaxed using production MD for 15 ns in the NPT ensemble at 310 K with the V-rescale thermostat and semi-isotropic Parrinello-Rahman pressure coupling. Time steps of 2 fs were used.

For each residue, 24 independent steered MD simulations were run for each pre-protein, where the substituted residue was pulled in a z-axis direction (up through the channel) using an umbrella

potential moving at a rate of 1 nm.ns$^{-1}$, with a force constant of 1000 kJ.mol$^{-1}$.nm$^{-2}$. For each repeat, the total pulling force (taken as the area under the curve) was recorded.

## Modelling of an engaged pre-protein in the SecA-SecYEG complex for pK$_a$ analysis

Initial coordinates for SecA-SecYEG were taken from chains A, C, D, and E of PDB 3DIN (*Zimmer et al., 2008*), with the ADP-BeF$_x$ molecule replaced with ATP (*Piggot et al., 2012*). Simulations were run over 1 µs in a POPC membrane with explicit waters and Na$^+$ and Cl$^-$ ions to 0.15 M using the OPLS-AA force field (*Jorgensen et al., 1996*), see *Allen et al., 2016* for full details. Taking a 1 µs snapshot as a fully equilibrated starting model, a region of pre-protein 76 residues long was built in an extended configuration through the SecA-SecYEG complex, as described previously (*Corey et al., 2016a*). The pre-protein was positioned such that it contacted previously identified crosslinking sites in both SecY and SecA (*Corey et al., 2016a*; *Park et al., 2014*). The N-terminal SS was modelled as a helix and sited in the SecY lateral gate a per cryo-EM density (*Park et al., 2014*).

The SecYEG-SecA-pOA-ATP model was then embedded in a POPC membrane, solvated with explicit waters and Na$^+$ and Cl$^-$ ions to 0.15 M, and subjected to 1 µs MD simulation. Simulations were carried out as previously described (*Allen et al., 2016*), using the OPLS-AA force field (*Jorgensen et al., 1996*), in Gromacs 5.0.4 (*Berendsen et al., 1995*).

## pK$_a$ scanning pipeline

To predict the p$K_a$ of charged residues as they traverse the SecA-SecYEG complex, we constructed a computational pipeline. For 20 different structural snapshots over the final 500 ns of the SecYEG-SecA-pOA simulations, each of the 76 residues in the pre-protein were substituted to lysine in turn, using Scwrl4 (*Krivov et al., 2009*), for a total of ca. 1500 snapshots. These were then relaxed for 1 ns using MD, and the p$K_a$ of the target lysine determined using propKa31 (*Søndergaard et al., 2011*).

## Construction of 1D free energy profiles

To provide a more detailed view of the energetic cost of transporting protonated lysine, we constructed free energy profiles of protonated and deprotonated lysine residues through the channel, using the Martini 2.2 force field (*Marrink et al., 2007*; *Monticelli et al., 2008*). Following conversion to Martini, elastic bonds of 1000 kJ mol$^{-1}$ nm$^{-2}$ were applied between all backbone beads within 1 nm. Electrostatics were described using the reaction field method, with a cut-off of 1.1 nm using a potential shift modifier, and van der Waals interactions were shifted from 0.9 to 1.2 nm. Simulations were run in the NPT ensemble, with V-rescale temperature coupling at 323 K and semi-isotropic Parrinello-Rahman pressure coupling.

We used steered MD to construct a 1D reaction coordinate for an Ala-Lys-Ala peptide through the SecY channel, with the collective variable constructed from the z-distance between each of the backbone beads in the tripeptide and 5 backbone beads forming the SecY pore (Met 75, Ile 78, Ile 183, Val 278 and Ser 401 in *G. thermodentrifinicans numbering*). We performed 200 ns umbrella sampling MD, using a z-axis umbrella force constant of 2000 kJ mol$^{-1}$ nm$^{-2}$, in 0.1 nm windows along this coordinate, with the lysine either protonated or deprotonated. Construction of the 1D free-energy profile was achieved for the last 150 ns of each window using the weighted histogram analysis method, implemented in the gmx wham program (*Hub et al., 2010*). Convergence was determined through analysis histogram overlap.

## Proton-motive force measurements

The PMF of *E. coli* whole cells was determined using the distribution of [14C]benzoate and [14C]methyltriphenylphosphonium$^+$ as previously described (*Rao et al., 2001*). Cultures were grown exactly as described for the induction of SecYEG, to allow for a better comparison between whole cells and the IMVs from which they are isolated, and adjusted to OD$_{600}$ = 1.0 using fresh expression media before measurement. The internal volume of these cells was estimated using the partitioning of $^3$H$_2$O and [14C]PEG-4000 as previously described (*Rao et al., 2001*). The PMF of IMVs was measured using [14C] methylamine and [14C]potassium isothiocyanate as previously described (*Reenstra et al., 1980*), with the following modifications: instead of flow dialysis, a vacuum manifold (Millipore) fitted with 0.45 µm HA MF membrane filters (Millipore) was used as previously described (*Gebhard et al., 2006*). Assays

were performed at a membrane concentration of 1 [mg protein]/mL in buffer (50 mM Tris-HCl pH 7.0, 5 mM $MgCl_2$, 100 mM KCl). 1 mM ATP was added and 2 minutes later cold 2 mL 0.1 M LiCl was used to stop the reaction and membranes were harvested by vacuum filtration, with a further wash of 2 mL 0.1 M LiCl. 250 nCi (4.5 µM and 4.2 µM of [$^{14}$C]methylamine and [$^{14}$C]potassium isothiocyanate respectively) was added per experiment. The interval volume of 1.09 µL [mg protein]$^{-1}$ has been previously calculated (*Reenstra et al., 1980*). 2 mL scintillation fluid (Amersham) was added to all samples in 4 mL scintillation vials and counted as previously described (*Gebhard et al., 2006*; *Rao et al., 2001*). Protein concentration was determined using the BCA assay (Thermo) using bovine serum albumin as a standard.

## Acknowledgements

We thank Gail Bartlett for help with the bioinformatics analysis. Extended simulations were run on the ARCHER UK National Supercomputing Service (http://www.archer.ac.uk), provided by HECBioSim, the UK High End Computing Consortium for Biomolecular Simulation (hecbiosim.ac.uk), supported by the EPSRC. For the purpose of Open Access, the authors have applied a CC BY public copyright licence to any Author Accepted Manuscript version arising from this submission.

## Additional information

### Funding

| Funder | Grant reference number | Author |
| --- | --- | --- |
| Wellcome Trust | 10.35802/104632 | William J Allen<br>Ian Collinson |
| Biotechnology and Biological Sciences Research Council | BB/S008349/1 | Daniel W Watkins<br>Ian Collinson |
| Biotechnology and Biological Sciences Research Council | BB/N015126/1 | Daniel W Watkins<br>Ian Collinson |
| Biotechnology and Biological Sciences Research Council | BB/M003604/1 | Robin A Corey<br>Ian Collinson |
| Biotechnology and Biological Sciences Research Council | BB/I008675/1 | William J Allen<br>Ian Collinson |

The funders had no role in study design, data collection and interpretation, or the decision to submit the work for publication. For the purpose of Open Access, the authors have applied a CC BY public copyright license to any Author Accepted Manuscript version arising from this submission.

### Author contributions

William J Allen, Conceptualization, Data curation, Formal analysis, Funding acquisition, Investigation, Methodology, Project administration, Validation, Writing – original draft, Writing – review and editing; Robin A Corey, Conceptualization, Data curation, Formal analysis, Investigation, Methodology, Software, Validation, Visualization, Writing – review and editing; Daniel W Watkins, Conceptualization, Investigation, Resources, Writing – review and editing; A Sofia F Oliveira, Formal analysis, Investigation, Methodology, Software, Writing – review and editing; Kiel Hards, Formal analysis, Investigation, Methodology, Validation, Visualization, Writing – review and editing; Gregory M Cook, Conceptualization, Formal analysis, Resources, Supervision, Writing – review and editing; Ian Collinson, Conceptualization, Formal analysis, Funding acquisition, Methodology, Project administration, Resources, Supervision, Writing – review and editing

### Author ORCIDs

William J Allen ⓘ http://orcid.org/0000-0002-9513-4786

Robin A Corey ⓘ http://orcid.org/0000-0003-1820-7993
Daniel W Watkins ⓘ http://orcid.org/0000-0003-3825-5036
A Sofia F Oliveira ⓘ http://orcid.org/0000-0001-8753-4950
Ian Collinson ⓘ http://orcid.org/0000-0002-3931-0503

### Decision letter and Author response

Decision letter https://doi.org/10.7554/eLife.77586.sa1
Author response https://doi.org/10.7554/eLife.77586.sa2

---

## Additional files

### Supplementary files

• Supplementary file 1. List of pre-protein sequences used. Differences from the native pSpy sequence are highlighted in bold. $pSpy_{LXX}$ is identical to $pSpy_{XLX}$, but with Pep86 and its linker (yellow and pink) moved in front of the variable pSpy (black)

• Supplementary file 2. Berkeley Madonna models.

• Supplementary file 3. Tables of best fits and p-values. **a: p-values for the difference in lag between each variant and wt.** Calculated using a two-tailed t-test (in Microsoft Excel). **b: List of best fit parameters for all $pSpy_{XLX}$ variants** Values and confidence intervals were estimated by fitting each experimental replicate individually (n = 12 for wt, n = 3 for all the others), and using the mean and SEM of the best fits values. **c: p-values for bioinformatic analysis of arginine/ lysine ratios.** Analyses performed using a two-tailed t-test in Microsoft Excel. **d: Comparison of the proton-motive force (PMF) generated by whole cells and inverted membrane vesicles (IMVs)**. Error bars indicate the standard deviation from three biological replicates (for cells) or three technical replicates (for IMVs). Raw data underlying the figures. For NanoLuc traces, separate runs are demarcated by alternating bold formatting.

• Transparent reporting form

### Data availability

All raw data generated during this study are included as supplementary files, and annotated with the figure they were used in.

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
