## [Editor Report]

Using a novel bioluminescence-based assay combined with rigorous kinetic modeling, Collinson and colleagues dissect the sequence features of client proteins that influence SecA/SecYEG-mediated protein translocation across the bacterial inner membranes. This study pushes the description of this important cellular pathway towards a highly detailed level, which will potentially advance our understanding of ATP-driven protein secretion mechanisms in bacteria.

---

## [Decision Letter]

**Decision letter after peer review:**

Thank you for submitting your article "Rate-limiting transport of positive charges through the Sec-machinery is integral to the mechanism of protein transport" for consideration by *eLife*. Your article has been reviewed by 3 peer reviewers, including Lejla Zubcevic as Reviewing Editor and Reviewer #1, and the evaluation has been overseen by Volker Dötsch as the Senior Editor. The following individual involved in the review of your submission has agreed to reveal their identity: James N Blaza (Reviewer #3).

Essential Revisions:

1) The reviewers agreed that the manuscript would benefit from having a more detailed explanation of the kinetic model in the manuscript. In particular, the kinetic model and fitting procedures are not clearly described. There are many terms that are not defined in the main text (such as the meanings of "the number of steps" [is it the number of kinetic steps described in Figure 1? or the number of amino acids that are transported upon ATP hydrolysis?], k_fail [the fail of which step in Figure 1?]), RMSD, "amplitude of variants" [is it the amplitude of the signal?], "best fit" [what is the criteria for the best fit?]).

In addition, the standard deviation (or error) of each fitted parameter is not reported. Also, the statistical significance of differences between different data (where it is relevant) is not reported either.

2) In p.5, "Therefore, to avoid detrimental changes of pre-protein behavior, we created constructs consisting of the pSpy SS followed by three tandem mature (m) Spys, with a pep86 sequence after the second …."

This statement is not supported by experimental data. That is, how did this construct design help avoid detrimental changes of preprotein behavior?

3) The deceleration of protein transport by Arg, the deprotonation of Lys during transport, and the effects of helicity (i.e., the local protein structure) on the transport are convincing. However, it was not considered how the sequence modification influences the global conformation of preproteins. For example, incorporating ~8 bulky hydrophobic residues such as Leu or Trp in place of moderately hydrophobic residues may induce the collapse of the Spy polypeptide chain at least the variable region and decelerate the transport. Please comment.

4) Related to 1b), authors employ the Kyte-Doolitle hydrophobicity scale (1982) to modify the hydrophobicity of preprotein. Several experimentally determined hydrophobicity scales have been developed that might be more sensible to use (e.g., Wimley-White, Hessa-von Heijne, Moon-Fleming, etc.). Although those scales show a reasonably good correlation with one another, the hydrophobicity of several residues are noticeably different in different scales. Please comment.

5) > p.6 lines 10-12: "We find that transport is slower (longer lag than native pSpyXLX above that of pSpyLXX, 0.65 min) when more bulky, hydrophobic and positively charged residues are present, while negative charges have limited effect."

versus

>p.6 lines 7-9: "Removing hydrophobic patches, meanwhile, has marginal if any effect on transport rate (Figure 2j), reaffirming that residue size is a bigger factor than hydrophobicity."

I strongly agree with the authors on the effects of Arg and Lys, but am not fully convinced with the effects of other features. In the former, I think that controlling the hydrophobicity by I>T and T>V effects is reasonable since the size differences between I, T and V are small. In F>W mutations, the newly incorporated bulkier H-bond forming aromatic residue can be involved in other through-space (nonlocal or local) interactions with the surrounding affecting the translocation rate. I wonder whether authors have tested W>F (inverse of F>W).

6) The result section Stimulation of transport by PMF is pH-dependent (p.13) is not conclusive.

The first paragraph: "To explore the effect of pH further we prepared IMVs at three different pHs (6.0, 7.0 and 8.0) and measured import of the pSpy4x series used previously (Allen et al., 2020) in the same buffer…."

Here, the rationale of the pH-dependence needs to be clarified. Also, it would be better to add a hypothesis derived from the experimental strategy and what are the expected results based on the hypothesis. At each pH, do readers expect that the pHs are the same between the inside and outside before turning on the proton pumping? Or, the designated pH represents only the outside of the IMV's before pumping?

7) The result section PMF stimulation of transport correlates with pre-protein hydrophobicity (p.14) is not conclusive.

This is an interesting result that the hydrophobicity bears a larger effect on the translocation rate rather than the charges. There is no satisfactory explanation and conclusion about this result. The two result sections overall weaken the impact of this manuscript although I am grateful for the authors' rigorous tests. I recommend that authors strengthen these two sections probably by emphasizing the explainable results and making the unexplainable results open questions.

8) Extending the conclusion to membrane protein integration.

In the abstract and discussion (the last paragraph), the authors extend the charge effect on SecA-mediated translocation to membrane protein translocation. The related statement would be convincing if preproteins contact the membrane during the passage through SecY channel. Based on the proposed model, it is not likely. Please comment.

9) The only major scientific query I have is to ask if the authors have tried measuring the change in δ-pH and deltaPsi in their IMVs in the presence of valinomycin and nigericin? At the moment it looks as if the measurement is only made on their preparations in the absence of these ionophores. For them to work there must be K^+^ in the lumen of the vesicle but there are reasons that the K^+^ may leak out after preparation (based on my own experiences with these systems). Given the exhaustive amount of data already in the paper, I would only ask the authors to comment either in reply or in the paper, there's quite enough experimental evidence already included.

---

## [Author Response]

Essential Revisions:1) The reviewers agreed that the manuscript would benefit from having a more detailed explanation of the kinetic model in the manuscript. In particular, the kinetic model and fitting procedures are not clearly described. There are many terms that are not defined in the main text (such as the meanings of "the number of steps" [is it the number of kinetic steps described in Figure 1? or the number of amino acids that are transported upon ATP hydrolysis?], k_fail [the fail of which step in Figure 1?]), RMSD, "amplitude of variants" [is it the amplitude of the signal?], "best fit" [what is the criteria for the best fit?]).

We have substantially expanded the description of the model in the main text, and defined all the terms in the model. Because of this all the absolute values have changed slightly, but all the conclusions remain the same.

These were originally omitted because they are defined extensively in a previous paper – but the reviewers are of course correct that this paper should be comprehensible on its own.

In addition, the standard deviation (or error) of each fitted parameter is not reported. Also, the statistical significance of differences between different data (where it is relevant) is not reported either.

We have refitted each NanoLuc data set individually, and used the averages of the individual fits (± SEM) to provide error estimates.

We have also carried out two-tailed t-tests to estimate statistical significance (where relevant). In general, pretty much every difference is very statistically significant as the NanoLuc traces are extremely precise and reproducible. So to avoid cluttering the manuscript we have mostly included these in supplementary tables (Supplementary tables 2-5).

2) In p.5, "Therefore, to avoid detrimental changes of pre-protein behavior, we created constructs consisting of the pSpy SS followed by three tandem mature (m) Spys, with a pep86 sequence after the second …."This statement is not supported by experimental data. That is, how did this construct design help avoid detrimental changes of preprotein behavior?

We have reworded this section to make it clearer. Essentially, we used the pSpy_XLX_ variants for two reasons: (1) to ensure that the initiation and end points of transport are identical for every variant, and (2) to avoid issues of solubility (and potentially affinity for SecA).

3) The deceleration of protein transport by Arg, the deprotonation of Lys during transport, and the effects of helicity (i.e., the local protein structure) on the transport are convincing. However, it was not considered how the sequence modification influences the global conformation of preproteins. For example, incorporating ~8 bulky hydrophobic residues such as Leu or Trp in place of moderately hydrophobic residues may induce the collapse of the Spy polypeptide chain at least the variable region and decelerate the transport. Please comment.

Collapse of hydrophobic regions does seem likely; this is one reason we spread the altered residues as evenly as possible through the sequence. Although if this affects our model pre-proteins, presumably the same effect should also affect real proteins, and so be part of the mechanism.

Fully distinguishing global and local effects is probably not possible using this assay setup. We have added some additional discussion of this point to the manuscript.

4) Related to 1b), authors employ the Kyte-Doolitle hydrophobicity scale (1982) to modify the hydrophobicity of preprotein. Several experimentally determined hydrophobicity scales have been developed that might be more sensible to use (e.g., Wimley-White, Hessa-von Heijne, Moon-Fleming, etc.). Although those scales show a reasonably good correlation with one another, the hydrophobicity of several residues are noticeably different in different scales. Please comment.

We chose Kyte-Doolittle simply because it seemed to be the most commonly used hydrophobicity scale, and had not appreciated that the various scales were significantly different.

Plotting the Wimley-White, Hessa-von Heijne and Moon-Fleming scales against Kyte-Doolittle (see new panels b-d in Figure 2 —figure supplement 2), it seems the biggest differences between scales are for the aromatic residues. In most cases, the substitutions we chose have the desired effect in all four scales. The only exception is the F→W mutant, in that according to the Wimley-White scale (but not the others) tryptophan it is more hydrophobic than phenylalanine, not less.

We have updated the text accordingly, and to make it clear that we are not fully able to distinguish hydrophobic from bulky residues.

5) > p.6 lines 10-12: "We find that transport is slower (longer lag than native pSpyXLX above that of pSpyLXX, 0.65 min) when more bulky, hydrophobic and positively charged residues are present, while negative charges have limited effect."versus>p.6 lines 7-9: "Removing hydrophobic patches, meanwhile, has marginal if any effect on transport rate (Figure 2j), reaffirming that residue size is a bigger factor than hydrophobicity."I strongly agree with the authors on the effects of Arg and Lys, but am not fully convinced with the effects of other features. In the former, I think that controlling the hydrophobicity by I>T and T>V effects is reasonable since the size differences between I, T and V are small. In F>W mutations, the newly incorporated bulkier H-bond forming aromatic residue can be involved in other through-space (nonlocal or local) interactions with the surrounding affecting the translocation rate. I wonder whether authors have tested W>F (inverse of F>W).

Unfortunately, the native Spy sequence has no tryptophans, so we were unable to try this substitution. In general, there is no obvious way to increase hydrophobicity while decreasing residue size for Spy (short of incorporating non-natural amino acids, which seems beyond the scope of this paper).

Overall, we have attempted to cover all general amino acid properties as thoroughly as possible, give the sequence constraints, and our conclusion is that transport time is dictated by (in order of importance):

arginine > residue size > hydrophobicity

We have made a few changes to this section and the discussion to make it clearer which conclusions are most firmly supported.

6) The result section Stimulation of transport by PMF is pH-dependent (p.13) is not conclusive.The first paragraph: "To explore the effect of pH further we prepared IMVs at three different pHs (6.0, 7.0 and 8.0) and measured import of the pSpy4x series used previously (Allen et al., 2020) in the same buffer…."Here, the rationale of the pH-dependence needs to be clarified. Also, it would be better to add a hypothesis derived from the experimental strategy and what are the expected results based on the hypothesis. At each pH, do readers expect that the pHs are the same between the inside and outside before turning on the proton pumping? Or, the designated pH represents only the outside of the IMV's before pumping?

We have completely rewritten this section as requested. It now explains the hypothesis properly, describes in more detail how the experiment is set up, and the conclusions the can reasonably be drawn.

7) The result section PMF stimulation of transport correlates with pre-protein hydrophobicity (p.14) is not conclusive.This is an interesting result that the hydrophobicity bears a larger effect on the translocation rate rather than the charges. There is no satisfactory explanation and conclusion about this result. The two result sections overall weaken the impact of this manuscript although I am grateful for the authors' rigorous tests. I recommend that authors strengthen these two sections probably by emphasizing the explainable results and making the unexplainable results open questions.

We agree that these results are not conclusive. They were included because – having performed the experiments – we felt it would be doing the field a disservice to omit them just because they didn't give the expected result.

We have rewritten this section and clarified that this is an open result for which we don't have a good explanation.

8) Extending the conclusion to membrane protein integration.In the abstract and discussion (the last paragraph), the authors extend the charge effect on SecA-mediated translocation to membrane protein translocation. The related statement would be convincing if preproteins contact the membrane during the passage through SecY channel. Based on the proposed model, it is not likely. Please comment.

What we were trying to say is that because the SecY channel is resistant to the passage of positive charges across the membrane during secretion, it is probably also resistant to the passage of positive charges on periplasmic loops of membrane proteins (which also pass through SecY). It is therefore possible that this might be part of the mechanism by which membrane protein topology is determined (the positive inside rule).

While this is clearly speculative, we believe it is an interesting enough notion to warrant inclusion in the discussion.

We have rewritten the final paragraph to make it clearer what we are referring to, and removed reference to it from the introduction.

9) The only major scientific query I have is to ask if the authors have tried measuring the change in δ-pH and deltaPsi in their IMVs in the presence of valinomycin and nigericin? At the moment it looks as if the measurement is only made on their preparations in the absence of these ionophores. For them to work there must be K^+^ in the lumen of the vesicle but there are reasons that the K^+^ may leak out after preparation (based on my own experiences with these systems). Given the exhaustive amount of data already in the paper, I would only ask the authors to comment either in reply or in the paper, there's quite enough experimental evidence already included.

We have not specifically measured ∆ψ and ∆pH in the presence of ionophores. Vesicles were prepared in the same buffer used for experiment, which contains 50 mM KCl. So we reasoned there is no opportunity for the K^+^ to leak out, and there should be plenty of K^+^ present for the ionophores to work.